# Effects of high spatial and temporal resolution Earth observation on simulated hydrometeorological variables in a cropland (southwestern France)

Jordi Etchanchu[1], Vincent Rivalland[1], Simon Gascoin[1], Jérôme Cros[1], Tiphaine Tallec[1], Aurore Brut[1], Gilles Boulet[1]

[1] CESBIO, Université de Toulouse, CNES, CNRS, IRD, UPS, Toulouse, France

*Correspondence to*: Jordi Etchanchu (jordi.etchanchu@cesbio.cnes.fr)

**Abstract.** Agricultural landscapes are often constituted of a patchwork of crop fields whose seasonal evolution is dependent on specific crop rotation patterns and phenologies. This temporal and spatial heterogeneity affects surface hydrometeorological processes and must be taken into account in simulations of land surface and distributed hydrological models. The Sentinel-2 mission allows for the monitoring of land cover and vegetation dynamics at unprecedented spatial resolutions and revisit frequencies (20 m and 5 days, respectively) that are fully compatible with such heterogeneous agricultural landscapes. Here, we evaluate the impact of Sentinel-2-like remote sensing data on the simulation of surface water and energy fluxes via the Interactions between the Surface Biosphere Atmosphere (ISBA) land surface model included in the EXternalized SURface (SURFEX) modelling platform. The study focuses on the effect of the Leaf Area Index (LAI) spatial and temporal variability on these fluxes. We compare the use of the LAI climatology from ECOCLIMAP-II, used by default in SURFEX-ISBA, and time series of LAI derived from the high-resolution Formosat-2 satellite data (8 m). The study area is an agricultural zone in southwestern France covering 576 km² (24x24 km). An innovative plot-scale approach is used, in which each computational unit has a homogeneous vegetation type. Evaluation of the simulations quality is done by comparing model outputs with in-situ eddy covariance measurements of latent heat flux (LE). Our results show that the use of LAI derived from high-resolution remote sensing significantly improves simulated evapotranspiration with respect to ECOCLIMAP-II, especially when the surface is covered with summer crops. The comparison with in-situ measurements shows an improvement of roughly 0.3 in the correlation coefficient and a decrease of around 30% of the Root-Mean Square Error in the simulated evapotranspiration. This finding is attributable to a better description of LAI evolution processes with Formosat-2 data, which further modify soil water content and drainage of soil reservoirs. Effects on annual drainage patterns remain small but significant, i.e., an increase roughly equivalent to 4% of annual precipitation levels with simulations using Formosat-2 data in comparison to the reference simulation values. This study illustrates the potential for the Sentinel-2 mission to better represent effects of crop management on water budgeting for large, anthropized river basins.

## 1 Introduction

In an agricultural river basin, farmer's practices have an impact on crop functioning. Farmers manage crop rotations, select variety, decide the seeding and harvest dates and organize irrigation supplements. In such basins, a more accurate description of crop dynamics and their effects on hydrometeorological fluxes is critical to improve the monitoring of water resources (Foley et al., 2005; Martin et al. 2016).

Land surface models (LSMs), such as the Variable Infiltration Capacity (VIC, Liang et al., 1994) or Interactions between the Surface Biosphere Atmosphere (ISBA, Noilhan and Planton, 1989) models, are increasingly used as distributed hydrological models to study and forecast water resource evolution (e.g., Habets et al., 2008; Tesemma et al., 2015). The Leaf Area Index (LAI) is defined as "half of the total developed area of green (i.e. photosynthetic active) leaves per unit ground horizontal surface area" (Chen and Black, 1992). It is the main variable used to parameterize the effect of vegetation dynamics on evapotranspiration in most LSMs. Garrigues et al. (2015) showed the importance of the LAI in evapotranspiration simulations. When it is not simulated by the model, the LAI is often derived or directly taken from reference tables organized by vegetation type (Verseghy et al., 1993; Maurer et al., 2002). This LAI is generally computed from low- to mid-resolution long-term satellite records and provided as a climatology. Thus it does not allow one to determine the impact of observed annual vegetation variability on water and energy fluxes on the land surface (Tang et al., 2012; Ford and Quiring, 2013). Studies have shown that prescribing a remotely sensed LAI with year-to-year variability in LSMs improves estimations of water and energy fluxes between the soil and atmosphere, mainly through a more realistic evapotranspiration (Van den Hurk et al., 2003; Jarlan et al., 2008). These studies used LAI values drawn from low- to mid-resolution satellite imagery, i.e., AVHRR (Van den Hurk et al., 2003) or MODIS (Tang et al., 2012; Ford and Quiring, 2013). However, for many cultivated areas and particularly in European countries (Fritz et al, 2015), field plot areas rarely exceed typical MODIS product pixel sizes (500 m, i.e., 25 hectares). As a result, MODIS pixels can contain mixed LAI signatures of different crop types with different phenologies. It thus degrades the actual temporal variability of the LAI on these fields. Consequently, it is not representative of the actual hydrometeorological behavior of each land cover type (Trezza et al., 2013; Nagler et al., 2013). This is particularly the case for regions where both summer and winter crops are cultivated, as in southwestern France. Indeed, summer and winter crops have anti-correlated phenologies so mixing these two LAI signatures leads to attenuating, or even suppressing of, the LAI variability throughout the year.

A potential solution to access realistic vegetation dynamic could be the use of high resolution remote sensing products. The recently launched Sentinel-2 mission generates multispectral imagery of land areas at a decametric resolution (10 m to 60 m depending on the band) over a 5-day revisit period with global coverage. Previous studies have already shown that higher resolution data can improve the description of the vegetation and modeled water processes in agricultural landscapes for which mid-resolution imagery is unsuitable (Ferrant et al., 2014; Ferrant et al., 2016). In this study, we used the Interactions between Surface Biosphere Atmosphere (ISBA) LSM as part of the EXternalized SURFace (SURFEX) modeling platform (Masson et al., 2013). SURFEX was developed by the French National Center for Meteorological Research (CNRM) to

represent and interface the surface processes for atmospheric and hydrological models. This platform is also used for research purposes in the fields of climatology, meteorology and hydrology. Within SURFEX, ISBA is the submodel in charge of simulating variables over natural emerged areas. It uses the ECOCLIMAP-II database to determine vegetation types and associated parameters (e.g., temporal LAI, fractional vegetation cover, albedo) at a spatial resolution of 1 km (Masson et al., 2003; Faroux et al., 2013). Each ECOCLIMAP-II grid cell is composed of up to 12 vegetation types or plant functional types (PFTs). Each PFT's LAI forcing in ECOCLIMAP-II database has a temporal resolution of 10 days. It is a climatology derived from MODIS satellite observations collected between 1999 and 2005. The LAI of each PFT is determined by unmixing the LAI of the MODIS grid cell using neighboring, unmixed pixels. The method, detailed in Faroux et al. (2013), consists of using the LAI of the nearest MODIS pixels with a pure PFT to unmix the effect of each PFT in the LAI signature.

Our study aims at evaluating the impact of introducing high-resolution information on vegetation type and LAI from Sentinel-2-like observations instead of the low resolution climatology of ECOCLIMAP-II in ISBA-SURFEX simulations. The main objective is to assess if this more accurate description of the phenological cycle, especially the agricultural practices mentioned above, translates into a better representation of the simulated evapotranspiration. We also aim at evaluating the impact on simulated drainage and runoff.

The study area is a pilot site in southwestern France (Dejoux et al., 2012). It is considered to be representative of the cultivated area in the upper Garonne River Basin. A fraction (estimated at 13%) of crop fields in the area is irrigated, but we chose not to focus on effects of irrigation due to a lack of spatially distributed data on irrigation quantity and timing. LAI and land cover maps were determined from a 5-years (2006-2010) time series of Formosat-2 satellite images, which has similar spectral and spatio-temporal characteristics as Sentinel-2. The LSM was applied at the "plot" scale to place it under homogeneous vegetation type conditions for each computation unit (one computation unit has only one PFT). This plot-scale modeling approach allows us to take into account of the spatial variability of LAI values between plots while limiting the computation time in comparison to a pixel-based approach. Our results are first compared to evapotranspiration in situ measurements. Then we perform a spatial comparison between simulation's results using the MODIS based ECOCLIMAP-II LAI forcing and the FORMOSAT-2 based LAI forcing. This way, we will point out the contribution on surface fluxes dynamics of using high spatial and temporal resolution vegetation forcing instead of a low resolution climatology. Finally, we discuss the limitations of this work and challenges that must be addressed in order to upscale this study to the Garonne River Basin scale using Sentinel-2 data.

## 2 Model and data

### 2.1 SURFEX-ISBA model and forcing

EXternalized SURFace (SURFEX) is a modeling platform developed by the CNRM/Meteo-France. It simulates exchanges between land surfaces and the atmosphere (Masson et al., 2013). It is composed of four modules to simulate radiative budget

and hydrological flux patterns for towns, lakes, oceans and natural areas. In this study, we used the Interactions between Surface Biosphere Atmosphere (ISBA, Noilhan et Planton, 1989) nature model included in the version 7-3 of SURFEX. The ISBA model uses meteorological and physiographic data to simulate energy and water fluxes between land surfaces and the atmosphere.

The version used in this study is the standard version of ISBA. It does not include a coupled stomatal conductance-photosynthesis scheme like in the A-gs version (Calvet et al., 1998). The water transfer in the soil is simulated on three layers with a force-restore approach presented by Deardorff (1977). This approach was integrated in ISBA by Mahfouf et Noilhan (1996). The three layers were described and calibrated by Boone et al. (1999). The surface layer volumetric water content is restored depending on the water content of both surface and root zone layer. A gravitational drainage flux is

simulated when the soil water content of a layer exceeds the field capacity. In the version we used, a subgrid runoff is also calculated using the Variable Infiltration Capacity scheme first described by Dumenil and Todini (1992) and included in SURFEX by Habets et al. (1999). It allows simulating a runoff flux even when the soil is not fully saturated. A unique energy budget is simulated for each vegetation type on the vegetation-soil layers composite by a single source scheme. A single surface temperature is used to compute the different energy fluxes. The energy budget calculation method is detailed

by Noihlan and Planton (1989).

The meteorological forcing for the study area is drawn from the Système d'Analyse Fournissant des Renseignements Atmosphériques à la Neige (SAFRAN, Durand et al. 1993) reanalysis data (Quintana et al., 2007), which provides precipitation data (in solid and liquid form), air temperature and specific humidity data at 2 meters, air pressure data, and wind and solar radiation data at hourly time intervals at an 8 km resolution. The SAFRAN data were spatially linearly

interpolated to match plot centroïds. Regarding soil parameters, by default, ISBA uses the Harmonized World Soil Database (FAO/IIASA/ISRIC/ISS-CAS/JRC, 2012), which gives the percentage of clay and sand at a 30 arc-second (~1 km) resolution. Soil parameters were then computed using empirical pedotransfer functions (Noilhan and Lacarrère, 1995).

The ECOCLIMAP-II database (Faroux et al., 2013) is used to describe land cover at a 1 km resolution, thus corresponding to the resolution of the SURFEX-ISBA simulation grid. This land cover database is divided in 273 ecosystems. These

25 ecosystems were determined by crossing a land cover classification derived from the SPOT/VEGETATION Normalized Difference Vegetation Index (NDVI) series and pre-existing land cover maps, as described by Faroux et al. (2013). Twelve different plant functional types (PFT, also referred to as patches) are considered to describe these ecosystems. Each ecosystem is described by a correspondant composition of these twelve patches. Each grid cell in ISBA belongs to a unique ecosystem and thus is described as a combination of these patches. Vegetation parameters are thus determined for each patch

of this ecosystem. Some of the parameters are fixed such as root depth and minimal stomatal resistance, and some are temporally variable such as albedo and LAI. In particular, the LAI follows a cycle determined from MODIS LAI analysis data for 2000 to 2005 averaged for each vegetation type of each ecosystem with a temporal resolution of 10 days. Because of the low spatial resolution of MODIS, the LAI signatures of several vegetation types are often mixed in a pixel. An unmixing method is then used by Faroux et al. (2013). It uses the nearest unmixed pixels of each PFT present in the MODIS pixel

considered to assess the contribution of each PFT in the LAI climatology. ISBA then uses these data to separately simulate all variables for each vegetation type present in the pixels, and then based on the fraction of each type as a weighting coefficient, it calculates global pixel fluxes.

By default, ISBA simulates the fluxes on a 1km regular grid corresponding to the ECOCLIMAP-II grid. But interpolation routines are included in it. It allows simulating on irregular grids, as done in this study.

## 2.2 Data

All remote sensing and in situ data were collected as part of the Observatoire Spatial Régional (OSR) project for an agricultural area of southwestern France near Toulouse (Fig. 1, Dejoux et al., 2012). This area is considered to be representative of the cultivated area of the Garonne River Basin, which is characterized by a variety of land cover forms. The two main types of crops found in this area are irrigated summer crops such as maize or soy plants and rain-fed rotation crops such as wheat and sunflower plants.

### 2.2.1 Formosat-2 Leaf Area Index

Formosat-2 is an NSPO (Taiwan) satellite that can generate daily multispectral images of the Earth's surface at an 8 m resolution and with a swath of 24 km. It functions on a tasking mode, i.e., it does not acquire data systematically like Sentinel-2 but rather must be programmed for a target area. Its sensor detects radiation within four frequency bands of blue, green, red and near-infrared. After geometric, atmospheric and radiometric corrections were made and clouds are detected (Hagolle et al., 2008 and 2010), measured reflectances were entered into the neural network BV-NET, which inverts the PROSAIL radiative transfer model (Claverie, 2012). This neural network deduces a set of vegetation parameters (among them the LAI and fraction of vegetation cover (FCOVER)) for each pixel. It thus generates 8-m resolution LAI maps for each date and pixels without cloud obstruction. Finally, we had access to 105 clear images on our study area for 2006-2010. The LAI product was validated by Veloso et al. (2012) for the same area and time period as those used in this study with destructive measurements on the vegetation. The time series of LAI maps was then spatially averaged at the plot scale using the land cover map (Sect. 2.2.2 below), was interpolated between available dates and was finally temporally averaged to obtain monthly forcings of LAI for each plot of the study area. The spatial averaging over each plot has been done with a 16m erosion of the plots (twice the size of a Formosat-2 pixel) to avoid border effects and impact of geo-location uncertainty (Sect. 5).

### 2.2.2 Formosat-2 land cover maps

Annual land cover maps were generated using the previously described Formosat-2 image time series (Fig. 2, Ducrot et al., 2005, 2007 and 2009). Then, a supervised classification algorithm based on Iterative Conditional Method (ICM) was applied to determine the vegetation type of each plot (Ducrot et al., 1998, Masse et al., 2011). This algorithm uses a learning sample composed of selected plots where the vegetation type is known. These plots are extracted from the "Politique Agricole

Commune" database, made of farmers' land use official declarations. The algorithm then uses the annual Normalized Difference Vegetation Index (NDVI) profiles of these plots to separate all the pixels into 34 classes with similar NDVI profiles.

### 2.2.3 In situ measurements

We used in situ measurements drawn from two eddy covariance stations in the study area located at Auradé (43°32'58.81" N, 01°06'22.08" E) and Lamasquère (43°50'05" N, 01°24'19" E) to evaluate the simulations. The Auradé plot is located on a hillside near Garonne river terraces. It belongs to a private cereal production farm with a wheat-sunflower-wheat-rapeseed rotation. At this site, only grain is exported, straw is stored and the plot is never irrigated. The Lamasquère plot is part of an experimental milk production farm. It is positioned along the Touch River and is characterized by a maize-winter wheat-maize-winter wheat rotation. All aboveground biomass is exported as cow feed and bedding. Maize grown in the Lamasquère plot is irrigated. In 2006, irrigation levels were measured at 147 mm between June and August.

Each flux site is equipped with 1) eddy covariance systems to measure half-hourly sensible heat flux and evapotranspiration, installed at 2.8 and 3.65 meters above the soil at Auradé and Lamasquère sites, respectively; 2) meteorological sensors to measure radiation (CNR1, Kipp & Zonen), wind speed (Windvane / prop Young), air temperature and humidity (HMP35, Vaisala); and 3) soil profile probes for water content measurements (CS616, Campbell Scientific) collected at depths of 5 cm, 10 cm, 30 cm, and 60 cm (and also 100cm at Lamasquère site). The eddy covariance (EC) system allows to monitor turbulent fluxes at fields scale combining synchronized measurements of 3-D wind components (Campbell, CSAT 3) and fluctuations of atmospheric $CO_2$ and $H_2O$ concentrations using a fast open path Infrared Gas Analyzer (LiCor LI-7500, IRGA). Evapotranspiration (ETR) and energy fluxes (latent heat LE and sensible heat H) are calculated and integrated over 30 minutes according to CarboEurope-IP recommendation (Aubinet et al., 1999, Béziat et al, 2009). Half-hourly fluxes were corrected for spectral frequency loss (Moore, 1986) and corrected for air density variations (Webb et al., 1980). Flux data were filtered and flagged according to statistics and objectives criteria: data out of range, rain event, friction velocity threshold, integral turbulence characteristic, stationarity test (Papale et al., 2006; Reichstein et al., 2005) and spatial representativeness (footprint) of the fluxes. For the latter, if the calculated fetch including 90 % of the flux (Kljun et al., 2004) model for each half-hourly EC flux value (*F-90*) was higher than the distance between the mast and the edge of the plot in the main wind direction, fluxes were discarded. Gapfilling was finally performed depending on the duration of missing data, either following the linear regression method (duration < 1h30), or following the mean diurnal variation or look up table method (duration >1h30) according to Beziat & al. (2009).

The LAI was also measured for these sites using a destructive method (Claverie 2012, Ferrant et al., 2014). At both flux sites, vegetation samples were collected along two transects crossing the field over the entire growing season until harvesting roughly once a month. Ten to twenty 1.5 m-long rows were collected on each sampling day. The organs of the collected plants were separated into green and yellow leaves, stems, flowers and fruits. The Plant Area Index (PAI) was defined as

half the surfaces of all green organs, and the Leaf Area Index (LAI) was defined as half the surfaces of green leaves; it was measured by means of a LiCor planimeter (LI3100, LiCor, Lincoln, NE, USA).

## 3 Methods

### 3.1 Numerical Experiments

We conducted two experiments to evaluate effects of the Formosat-2 LAI and land cover maps on ISBA simulations. The study area covers a 24x24 km area near Toulouse in southwestern France (Fig. 1). Simulations were carried out from 2006 to 2010. Our objective was to preserve the uniqueness of vegetation types within computation units to avoid mixing the LAI profiles of several crop types. A discretization of the area with a regular grid based on cartographic coordinates as it is done by default in ISBA would require employing a grid resolution of at least 50 m to capture the spatial heterogeneity of the

landscape (230,400 grid cells). The study area was thus discretized using an original approach: rather than using a regular grid, we used the land cover map to identify connected regions of pixels sharing a common PFT (using GDAL polygonize utility with 4-pixel connectedness). This discretization does not necessarily match actual crop fields because two adjacent plots with the same PFT are merged into one plot. However, in general, a "numerical plot" corresponds to a cultivated plot. A plot can also correspond to an uncultivated area, such as a forest patch. These homogeneous plots were determined for

each year of the simulation period, as the land cover maps differ from one year to another mainly due to crop rotations. The plot approach generates lower computation costs than the regular grid approach. The study area is composed of 12,500 to 14,500 plots depending on the year considered, representing 84% to 91% of the total image area. The remaining surface corresponds to roads, lanes, rivers and strips of lawn between fields, which are not simulated in this study.

The first experiment (ECOCLIMAP) is the reference simulation. It simulates fluxes based on ECOCLIMAP-II vegetation

parameters, including the LAI climatology and vegetation fraction (Sect. 2.1). ECOCLIMAP parameters are interpolated on plot centroïds with interpolation functions included in ISBA.

The second experiment (FORMOSAT) was carried out by prescribing the LAI using the monthly Formosat-2 LAI (Sect. 2.2.1) rather than the ECOCLIMAP-II LAI. Each plot was also assigned a unique PFT obtained from the FORMOSAT land cover maps. We first aggregated the 34 classes of the original land cover maps to match the 12 standard PFTs of SURFEX

(Table 1). The other vegetation parameters were drawn from ECOCLIMAP-II for the corresponding PFT.

### 3.2 Comparison methods

First, we did a local comparison with in-situ measurements. We extracted the outputs of both simulations from the Auradé and Lamasquère station plots. We then calculated correlation coefficient ($R^2$) and Root-Mean Square Error (RMSE) values between monthly cumulated measured and simulated evapotranspiration fluxes (ET).

Then, we did a spatialized comparison over the entire study area. We analyzed differences between both simulations for the entire study area by calculating correlation coefficients between the monthly simulated evapotranspiration time series for

each plot. These correlation maps allow one to identify plots where effects of using Formosat-2 data on the temporal evolution of evapotranspiration are more pronounced. Eventually, we aggregated the simulated LAI, evapotranspiration, drainage, runoff and Soil Water Index (SWI, Eq. (1), Le Moigne 2012) values of all of the plots based on PFT values to analyze effects of the Formosat-2 products by vegetation type. Each plot has a unique ISBA patch in the FORMOSAT experiment, forced by the land cover map. Thus only the corresponding patch was taken into account when comparing with the ECOCLIMAP experiment. If the corresponding patch was not present in the combination of patches given by ECOCLIMAP-II for the plot, then this plot was excluded of the results. By this way we are sure that we can compare the fluxes on specific vegetation types.

$$SWI = \frac{w - w_{wilt}}{w_{fc} - w_{wilt}} \tag{1}$$

where w is the volumetric soil water content, $w_{wilt}$ is the volumetric soil water content at the wilting point and $w_{fc}$ is the volumetric soil water content at field capacity.

## 4 Results

### 4.1 Local comparisons with in situ measurements

First, the simulated ET of both experiments has been compared to the measured ET on the study sites Auradé and Lamasquère. It shows that using LAI derived from Formosat-2 data in the SURFEX simulation improves the correlation and RMSE of almost every year with respect to the ECOCLIMAP experiment (Tables 2 & 3). The data for the years 2008 and 2010 at the Auradé site are not presented due to technical problems on the eddy-covariance tower, which affects fluxes estimations on these periods. The improvement is more significant when measurement fields are covered in sunflower or maize crops, i.e. summer crops, with an improvement of the R² of roughly 0.3 and a decrease of RMSE by around 30% . By contrast, the effects of Formosat-2 data are not as strong for wheat and rapeseed crops, i.e. winter crops, where the improvement on R² rarely exceeds 0.1, the decrease in RMSE being around 20%.

To understand why these differences appear, we compared the time series of measured and simulated LAI and evapotranspiration for both sites in 2006 (Fig. 3 & 4). Silage maize was grown in Lamasquère in 2006. This crop is harvested before plant senescence. Hence, the observed cycle is shorter than a typical maize cycle. Figure 3 shows that the LAI is more realistic when using Formosat-2 data rather than the ECOCLIMAP-II MODIS climatologic LAI. The FORMOSAT LAI phenological cycle is shorter. It is in agreement with the LAI cycle observed in situ. LAI value increase due to crop growth and sudden drop due to harvesting are well represented. By contrast, the ECOCLIMAP LAI is too high in the autumn and winter.

This better description of the LAI based on Formosat-2 data leads to a better simulation of evapotranspiration timing (Fig. 4). In particular, the evapotranspiration peak is delayed by one month for summer crops (i.e., on Lamasquère). It thus fits the measurements better. However, the FORMOSAT experiment results do not match the actual amplitude of the measured evapotranspiration peak. This difference will be discussed in the discussion part (Sect. 5).

By contrast, differences in evapotranspiration levels are minor for winter wheat at the Auradé site. LAI dynamics are similar. The main difference between observation and simulations for both experiments occurs after the harvest period but does not lead to large differences in ET values (Sect. 4.2).

Analyzing the results over the entire period (2006-2010) for both sites (Fig. 5 & 6) confirms what was observed in 2006. Indeed, the effect remains small on winter crops (Auradé: 2006 & 2009, Lamasquère: 2007 & 2009) but the delay in the

10 evapotranspiration peak is clearly visible on summer crops (Auradé: 2007, Lamasquère: 2006, 2008 & 2010). The underestimation of the simulations is also visible for the Lamasquère site on 2008 and 2010 but not as marked in 2008. This point will also be discussed in the discussion part (Sect. 5).

## 4.2 Spatial comparisons with the Formosat-2 image

We computed the correlation coefficient between the simulated evapotranspiration time series for both experiments for each

15 plot of the study area. Figure 7 illustrates the distribution of correlation coefficients grouped by land cover type. A small correlation value denotes that the evapotranspiration time series are not in phase. Evapotranspiration patterns are not heavily modified outside of the crop fields (Fig. 7a). It generates a correlation coefficient of almost 1 and low levels of value dispersion. From the crop areas, two populations can be identified: winter and summer crops. The temporal evolution of winter crop evapotranspiration (mostly wheat, C3) is not heavily modified (Fig. 7b). However, the effect is much more

significant for summer crops (mostly maize (C4), sunflower (C3) and soy (C3) plants). In this case the median correlation coefficient is lower than that of wheat. The degree of value dispersion is also considerable depending on the year (Fig. 7c & 7d).

To further understand effects on hydrometeorological processes, we compared the monthly differences between both experiments on LAI and evapotranspiration (ET) dynamics. The results were aggregated by averaging each variable of all of

25 the C4 crop fields (maize and sorghum) for 2008 (Fig. 8a). We also compared daily Soil Water Index (SWI, Fig. 8b), drainage (DRAIN) and runoff (Fig. 8c) differences. As observed previously (Sect. 4.1), the difference in ET denotes the delay in the evapotranspiration peak with a negative difference occurring during the spring (ECOCLIMAP is higher than FORMOSAT) and with a peak occurring during the summer (Fig. 8a). It is strongly correlated with the difference in LAI. The lower LAI level occurring during the spring in the FORMOSAT experiment induces a lower transpiration level because

the evaporative surface is restricted, thus enhancing stomatal resistance. As a result, the SWI in the FORMOSAT experiment remains higher than that of the ECOCLIMAP experiment until the summer (Fig. 8b). Then more water remains available for evapotranspiration during summer. Consequently the ET is higher during this period according to FORMOSAT experimental results. This explains why the ET difference is positive during the summer even when there is almost no

difference in LAI values between the experiments. In ISBA, drainage only occurs when the SWI is higher than 1 (Le Moigne, 2012). Hence, an increase in the SWI during the spring causes an increase in the drainage volume (Fig. 8c). This increase can be significant over the year (Table 4), averaging at approximately 4% of annual precipitation and at up to 8% for sunflower and wheat crop fields. Runoff is also affected by differences in the SWI (Table 5) but in smaller proportions.

Indeed, the function used in both of our experiments to simulate runoff is based on the premise that even when the SWI is lower than 1, rain can saturate part of a pixel's upper soil layer, thus generating runoff in the pixel. This dripping section of a pixel increases with soil moisture. This is not the case for drainage patterns, which require a saturation of the entire soil root and sub-root layers to occur. Thus, a higher SWI, even if it remains below a value of 1, suggests that a larger part of a pixel is dripping, thus causing runoff levels to be higher in the FORMOSAT experiment (Fig. 8c).

**5 Discussion**

**5.1 Uncertainties on remote sensing data**

Thanks to the high spatial resolution of Formosat-2, the plot scale modeling approach could be applied at regional scale. It allows distinguishing the effects on specific vegetation types in the context of LSM studies. The revisit frequency is also sufficient to clearly monitor the phenological cycle and its critical stages at the plot scale. The local comparison between

model and measurements (Sect. 4.1) clearly shows that the use of Formosat-2 (Sentinel-2 like) data allows the model to capture the seeding and harvest dates (Fig. 3 & 5) unlike with the ECOCLIMAP-II forcing. Note that the quality of the Formosat-2 acquisition is quite similar to the Sentinel-2's (Koetz et al., 2017). Especially, the geo-location uncertainty is smaller than the size of a pixel, what allows precise extraction of LAI values from plots geometry. We thus assume that it has no impact given the erosion of the polygons within our LAI retrieval method (Sect. 2.2.1). The uncertainties on

radiometry are also rather small compared to the measurements' uncertainties while the Signal-Noise Ratio (SNR) is satisfying. The inner-field variability (green area, Fig. 3) is also clearly smaller than the measurements' uncertainties. However, some differences between the measured and remotely sensed LAI can be found. Particularly, the maximum LAI in Formosat-2 data tends to be lower than the measured one. This is due to a saturation effect of the remotely sensed LAI, pointed out by Veloso et al. (2012). But the impact on the simulated evapotranspiration amplitude is not significant as shown

by the example of a winter wheat field (Fig. 3a & 4a). The growth and senescence periods might also be inaccurate on some plots of the area, or even completely missing, because of the cloud obstruction. Indeed, cloud coverage is the main limitation of the use of optical remote sensing products. Even with such a high revisit frequency, a plot can remain obstructed over all the revisit dates during the phenologic cycle. The Auradé site in 2008 shows a typical example of this phenomenon (Fig. 5).It means that for some regions of the world, like tropical regions where the cloud coverage is frequent, this method would

not be appropriate.

## 5.2 Impact of remote sensing data on simulated evapotranspiration

The high-resolution LAI forcing has modified the simulated evapotranspiration, giving a more realistic temporal dynamics. The uncertainties on eddy-covariance measurements (gray area on fig. 4) are calculated from the frequency response correction uncertainty, the Webb-correction (turbulent environment) uncertainty, the calibration correction uncertainty and the random uncertainty, following Kroon et al. (2010). Total uncertainty is proportional to the flux itself and therefore uncertainty grows with the evapotranspiration flux. However, these uncertainties remain very small, representing barely 5% of the flux value during summer. Another approach to evaluate the uncertainty is the verification of the energy budget closure. P. Béziat (2009) evaluated the energy budget closure on both sites for the period 2005-2007. His conclusion is that the uncertainty related is very acceptable, the energy budget being closed at around 85-90%. The best results are obtained during the crop cycle. The difference between the evapotranspiration simulated in the ECOCLIMAP experiment and the measurements shows relative error exceeding 100% in spring on summer crops (Fig. 4 & 6). Thus, it exceeds by far the relative uncertainty expected on the eddy-covariance measurements even while taking the energy budget closure uncertainty into account.

The results (Sect. 4) show that the effect is more significant on specific crops, i.e. summer crops like maize or sunflower.. The impact on evapotranspiration seems to be quite similar for all summer crops fields. But the causes of this impact differ from a cultivation type to another. The case of the maize crops (Fig. 3b & 8a) shows the limits of the ECOCLIMAP-II unmixing algorithm of MODIS LAI data. In the case of C4 crops, the ECOCLIMAP-II LAI remains far too high during winter and growth and senescence periods. The resulting fluxes are thus particularly affected (Fig. 4b & 8a). It may be attributable to (i) the unmixing algorithm itself, (ii) the temporal averaging method used to create the LAI climatology, and (iii) small woodland areas or strips of lawn between crops fields that cannot be resolved using MODIS, although they maintain moderate LAI values during the winter. The case of sunflower and soya crops (Fig. 7b) is slightly different. These plants are considered as C3 crops like wheat. But the ECOCLIMAP-II phenology for C3 crops is mainly that of wheat in this region. Consequently, the sunflower is always simulated as wheat whereas their phenologies and hydrometeorological behaviors are very different. Replacing the C3 and C4 classes of ECOCLIMAP-II by a more detailed classification separating summer and winter crops could be of great interest for Land Surface modeling. Even with a MODIS LAI climatology based on the same unmixing algorithm, it could increase the precision of the simulated evapotranspiration especially for the summer crops. The interannual variability of the results on evapotranspiration (Fig. 7) may be justified by the climatic conditions of each year. Indeed, climatic conditions influence the farmers' decisions concerning the seeding and/or harvest dates. If these dates are closer than the ones simulated by ECOCLIMAP LAI, the effect on evapotranspiration is weaker.

## 5.3 Limitations and perspectives

Even if the evapotranspiration dynamic is more realistic when using Formosat-2 products, the model remains unable to simulate the actual amplitude of the measured evapotranspiration flux (Fig. 4b). This is most likely due to the fact that the model does not simulate irrigation while the Lamasquère site was irrigated between June and August 2006 (147 mm). The same conclusion applies for 2010 for the Lamasquère site (Fig. 6). Concerning 2008, also for Lamasquère, the precipitation amount was sufficient to limit the irrigation. Hence the amplitude of the measured evapotranspiration is lower. Thus the lack of irrigation in the model does not lead to a big difference with the measurements on the maximal amplitude of the evapotranspition, in contrast with 2006 and 2010. Outside of the irrigation period, as well as on rain fed plots, the simulated evapotranspiration when the model is forced by the Formosat-2 appears really close to the measured evapotranspiration (Fig. 4 & 6). The discrepancy between the simulations and the measurments is larger than the observation uncertainties as computed with the method of Kroon et al. (2010, Fig. 4). However, by considering the energy balance closure approach to estimate the observation errors (Beziat, 2009), the difference between the model and the observations is significant only during irrigation periods (Fig. 6). Adding the measured irrigation rates to the SAFRAN precipitation forcing improves this simulation with respect to ET measurements (not shown here). Ignoring irrigation remains the main limitation of our study given that roughly 13% of the plots in our study area are irrigated. Including irrigation will improve the simulations (Garrigues et al., 2015). An automated irrigation module might therefore be a significant improvement even if it often relies on poorly known soil parameters, like the available water content for evapotranspiration depending on the field capacity, the wilting point and the root depth. Incorporating irrigation rules similar to those of the MAELIA platform (Therond et al., 2014) could add up to the differences observed in this study and give even better results. Adding a new set of spatialized soil parameters, to pilot more efficiently the maximum available water content of the soil, could be of great interest. These issues will be addressed in the future. The spatialized comparison also shows that the soil water content, the drainage and the runoff are significantly increased (Fig. 8b & 8c) especially on summer crops. Such a difference in these fluxes could have an impact on the river discharge and groundwater recharge. Hence, further work is necessary to upscale this study at the scale of a river basin, which is the scale of the water management agencies. Sentinel-2A already images the Earth' land area at a similar radiometric spatial resolution than Formosat-2 over a 10-day revisit period. The optimal revisit period of 5 days should be achieved after the launch of Sentinel-2B. Scalable algorithms for crop type mapping and LAI retrieval are now available (Li et al., 2015; Inglada et al., 2015), allowing for the processing of large areas. While the computation costs of the hydrometeorological model at this spatial resolution may be an issue, the SURFEX modeling platform can run in a parallel mode using the Message Passing Interface protocol. These advances could allow studying the impact on the river discharge.

## 6 Conclusion

This study deals with the inter-comparison between two spatialized hydrometeorological ISBA modeling approaches in an agricultural zone in southwestern France from years 2006 to 2010. A first experiment was performed with LAI forcing from

ECOCLIMAP-II database which was generated from MODIS data. It was considered as the reference simulation. Second experiment included LAI forcing from high resolution Formosat-2 (Sentinel-2 like) time series data. Both simulations were performed with plots as computing units, where plot segmentation was derived from Formosat-2 high resolution land cover maps classifications. The use of the plot scale approach allowed exploiting the high spatial resolution on coherent

hydrometeorological units while limiting the calculation time compared to a pixel based approach. Thanks to the high revisit frequency of Formosat-2, the complex anthropogenic effects which affect land surface properties (e.g., seeding and harvest dates, crop rotations) can be captured. The comparison between the two experiments reveals significant differences in the simulated water fluxes. The results shows that summer crops LAI dynamics appear more realistic when using Formosat-2 data. Consequently, the modeled evapotranspiration also appears more realistic on this kind of crop. These results point out

the limitations of both the LAI retrieval method of ECOCLIMAP-II and the lack of inter-annual variability of the vegetation in the model. As expected, however, the incorporation of satellite LAI was not sufficient to capture the amplitude of the evapotranspiration peak in the validation site where irrigation is practiced. Indeed, there is no parameterization for irrigation practices in our model while the irrigated area is known to be 13% of plots in the study area. Hence, the focus will now be put on the representation of the irrigation in the model. This will allow a further evaluation of the model at the catchment

scale based on the observed river discharge.

**The supplements related to this article are available online at: [http://tully.ups-tlse.fr/simon/surfex-configuration-files](http://tully.ups-tlse.fr/simon/surfex-configuration-files)**

**Code and data availability**

The version of SURFEX used in this study is the v7.3. The code is available here: [http://www.umr-cnrm.fr/surfex//data/BROWSER/out_doc73/index.html](http://www.umr-cnrm.fr/surfex//data/BROWSER/out_doc73/index.html).

For the LAI data or land cover maps, please ask to the corresponding author.

**Author contribution:**

J. Etchanchu formatted the data, performed the simulations, analyzed the results and wrote most of the current paper.

25    V. Rivalland and S. Gascoin helped at designing the experiments, processing the data and analyzing the results.

J. Cros provided the tools to process the remote sensing data.

A. Brut wrote the section 2.2.3 about the instrumentation of the two stations.

T.Tallec calculated and provided the associated uncertainties of in-situ measurement.

All the authors revised the paper.

**Acknowledgments:**

This study was conducted as part of the REGARD project entitled « Modélisation des REssources en eau sur le basin de la GAronne, interaction entre les composantes naturelles et anthropiques et apport de la téléDétection » (http://www.cnrm-game-meteo.fr/spip.php?article809) funded by the French "Sciences and Technologies for Aeronautics and Space" Foundation (http://www.fondationstae.net). We thank Mathieu Coustau for his work, which formed the basis of our paper. Our sincere thanks go to Claire Marais-Sicre who provided the Formosat-2 land cover maps. We thank all the people of the OSR who contributed to the collect and the process of data. OSR facilities and staff are funded and supported by the Observatory Midi-Pyrenean, the University Paul Sabatier, Toulouse, France, CNRS (Centre National de la Recherche Scientifique), CNES (Centre National d'Etude Spatial) and IRD (Institut de Recherche pour le Développement). We especially thank the European Research Infrastructure Consortium Integrated Carbon Observation System (ERIC ICOS) and ICOS-France for funding facilities and staff working on Auradé and Lamasquère sites.

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

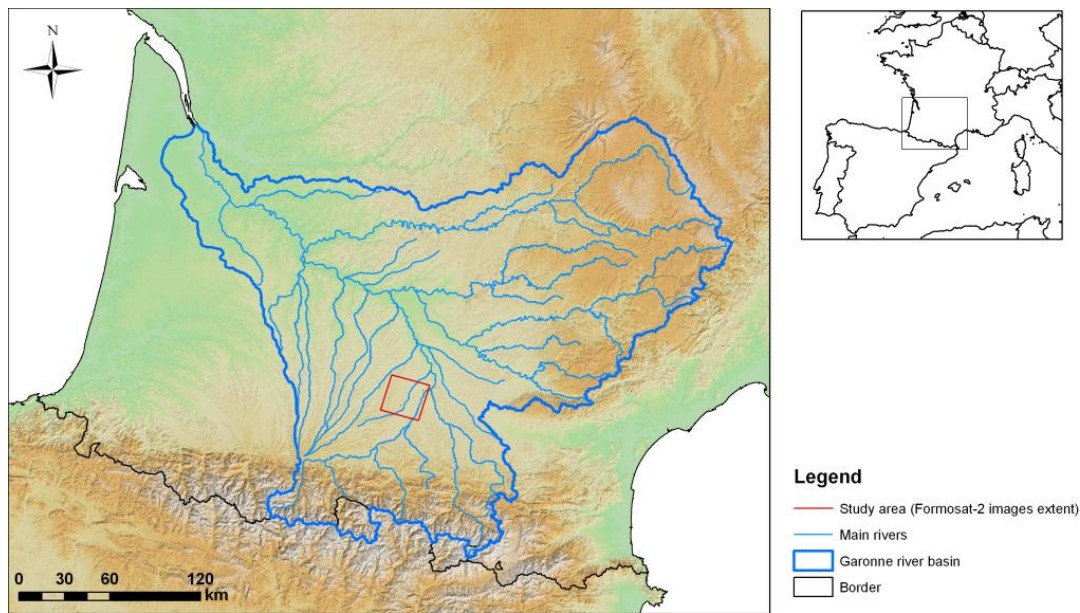

**Figure 1: Study area location (red).**

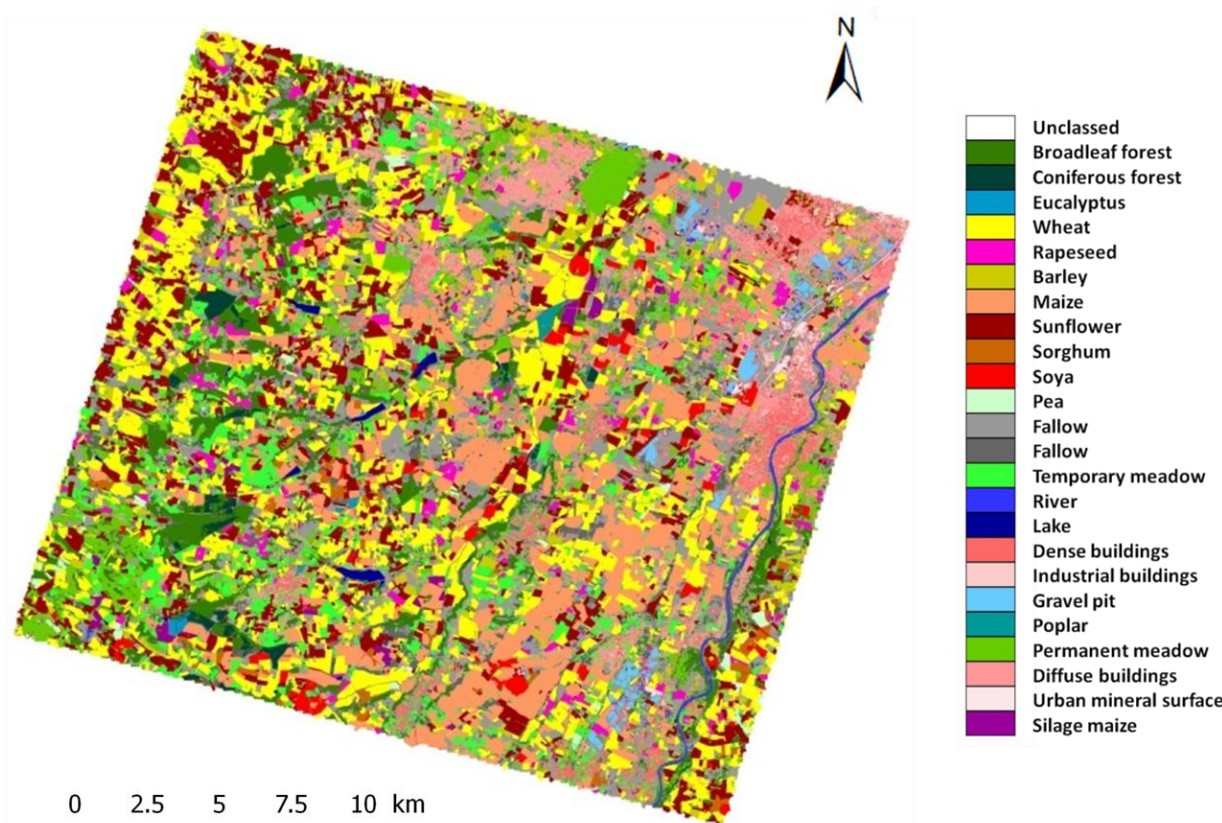

**Figure 2: Land cover map for 2006.**

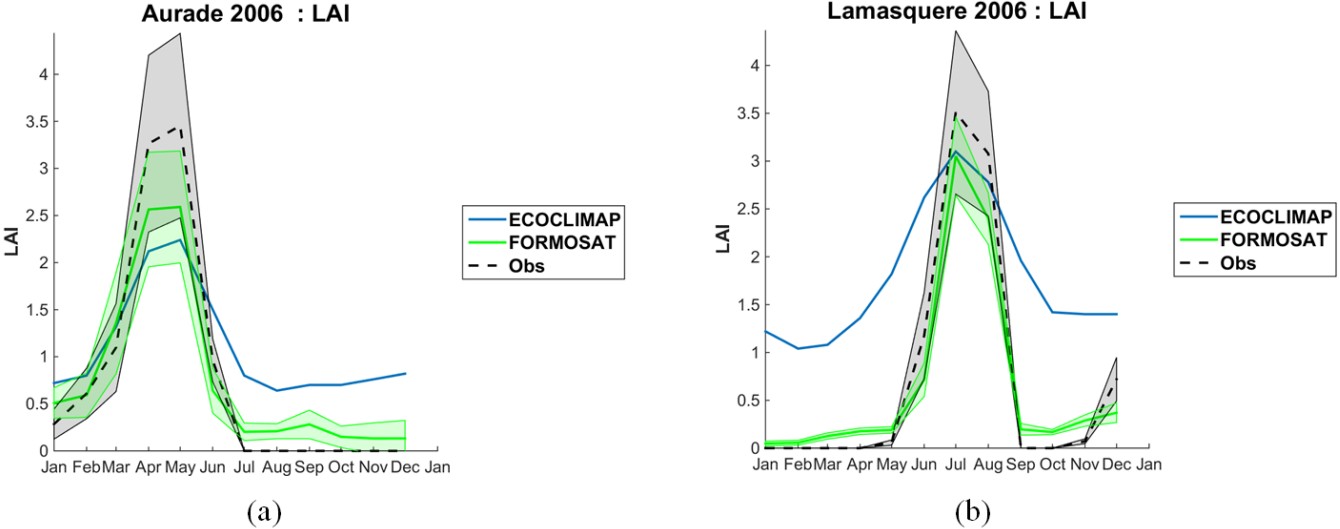

(a)                                                        (b)

**Figure 3: LAI time series for Auradé (a) and Lamasquère (b) for 2006. The filled areas represent the standard deviations for the measured (gray) and remotely sensed (green) time series.**

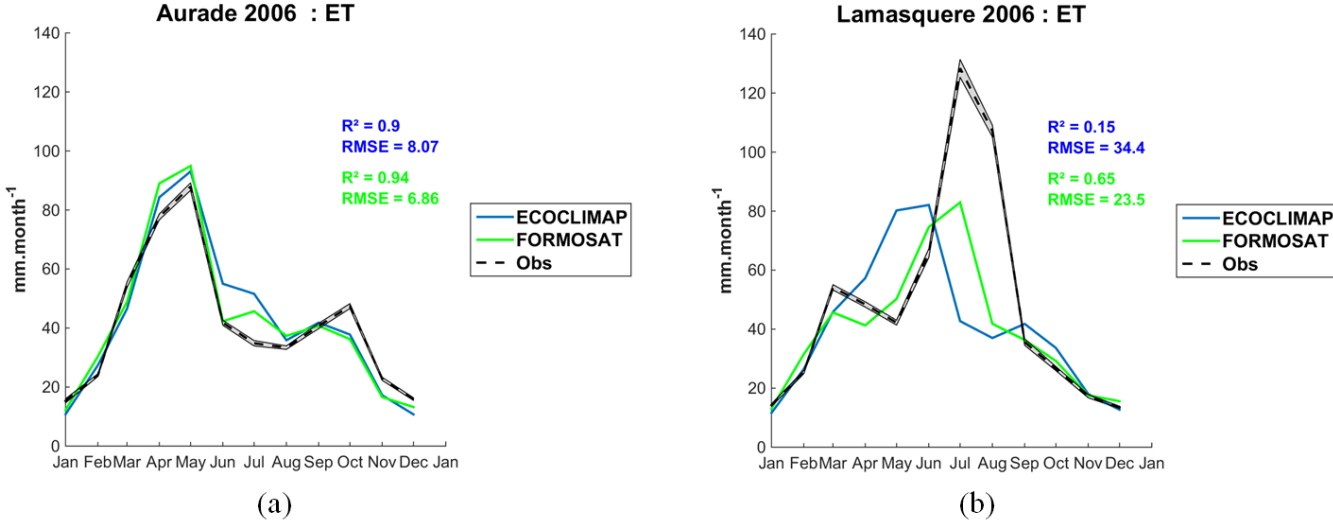

(a)                                                        (b)

**Figure 4: Evapotranspiration time series for Auradé (a) and Lamasquère (b) for 2006. The gray filled areas represent the standard deviations for the measured time series.**

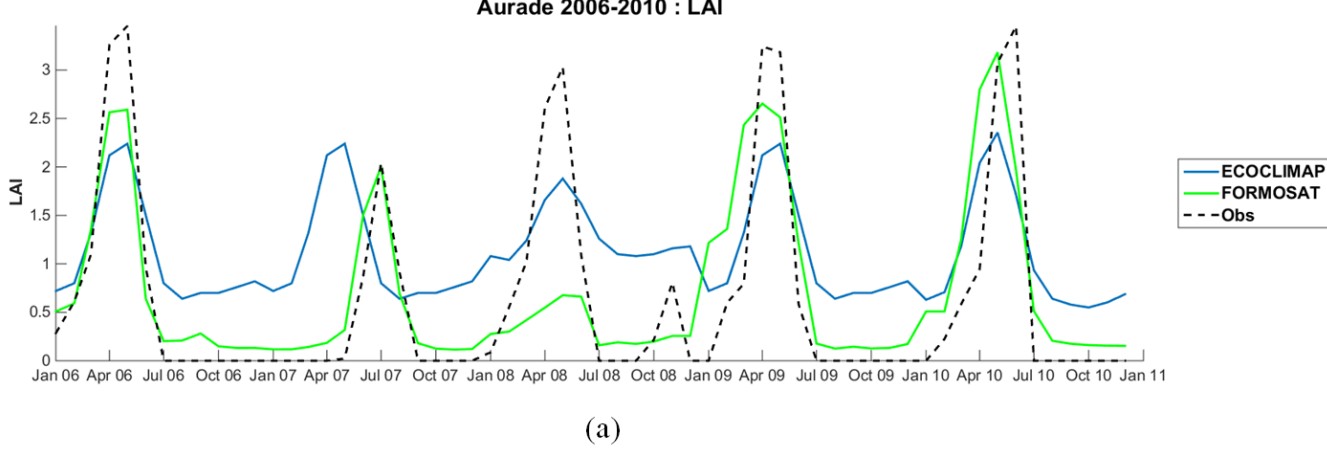

(a)

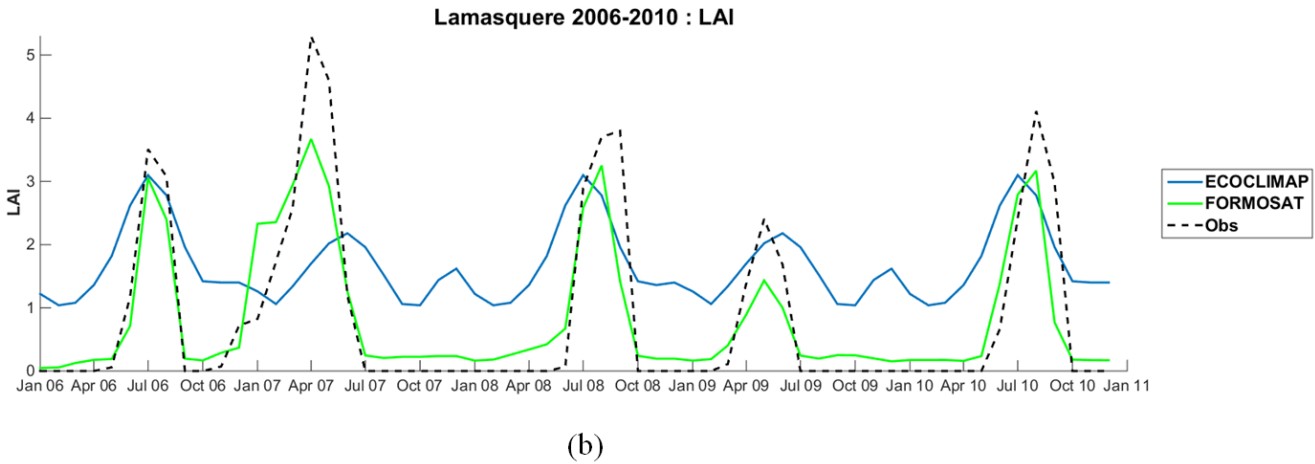

(b)

**Figure 5: LAI time series for Auradé (a) and Lamasquère (b) for 2006 to 2010. The crop type for each year is described in tables 2 and 3.**

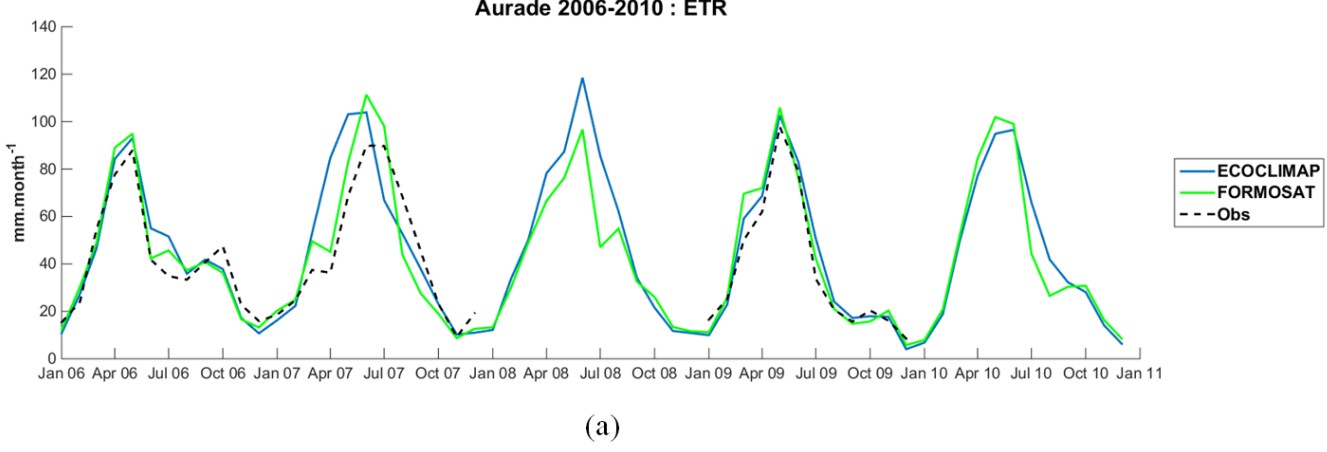

(a)

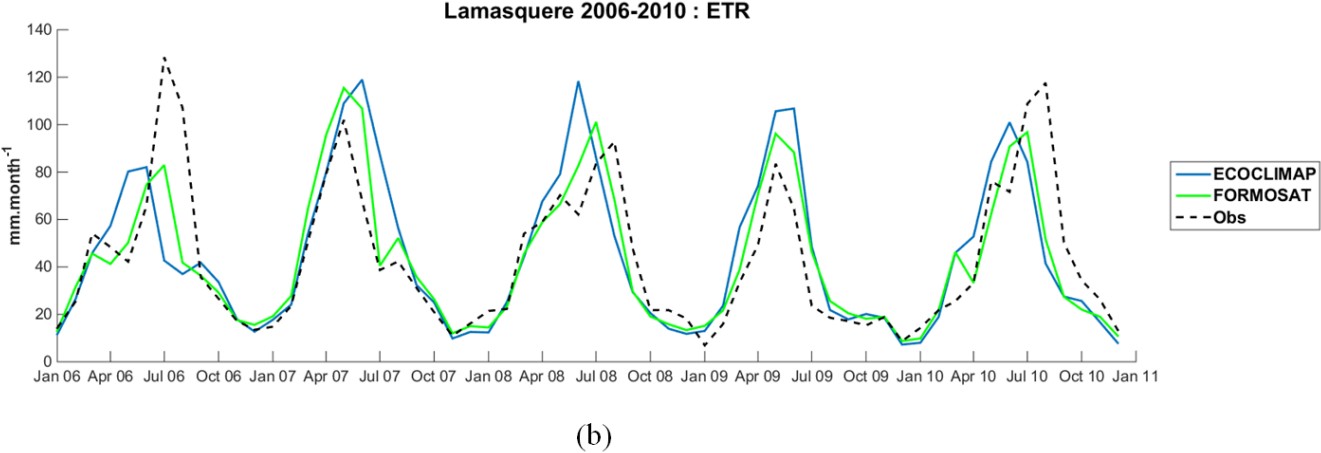

(b)

**Figure 6: Evapotranspiration time series for Auradé (a) and Lamasquère (b) for 2006 to 2010. The crop type for each year is described in tables 2 and 3.**

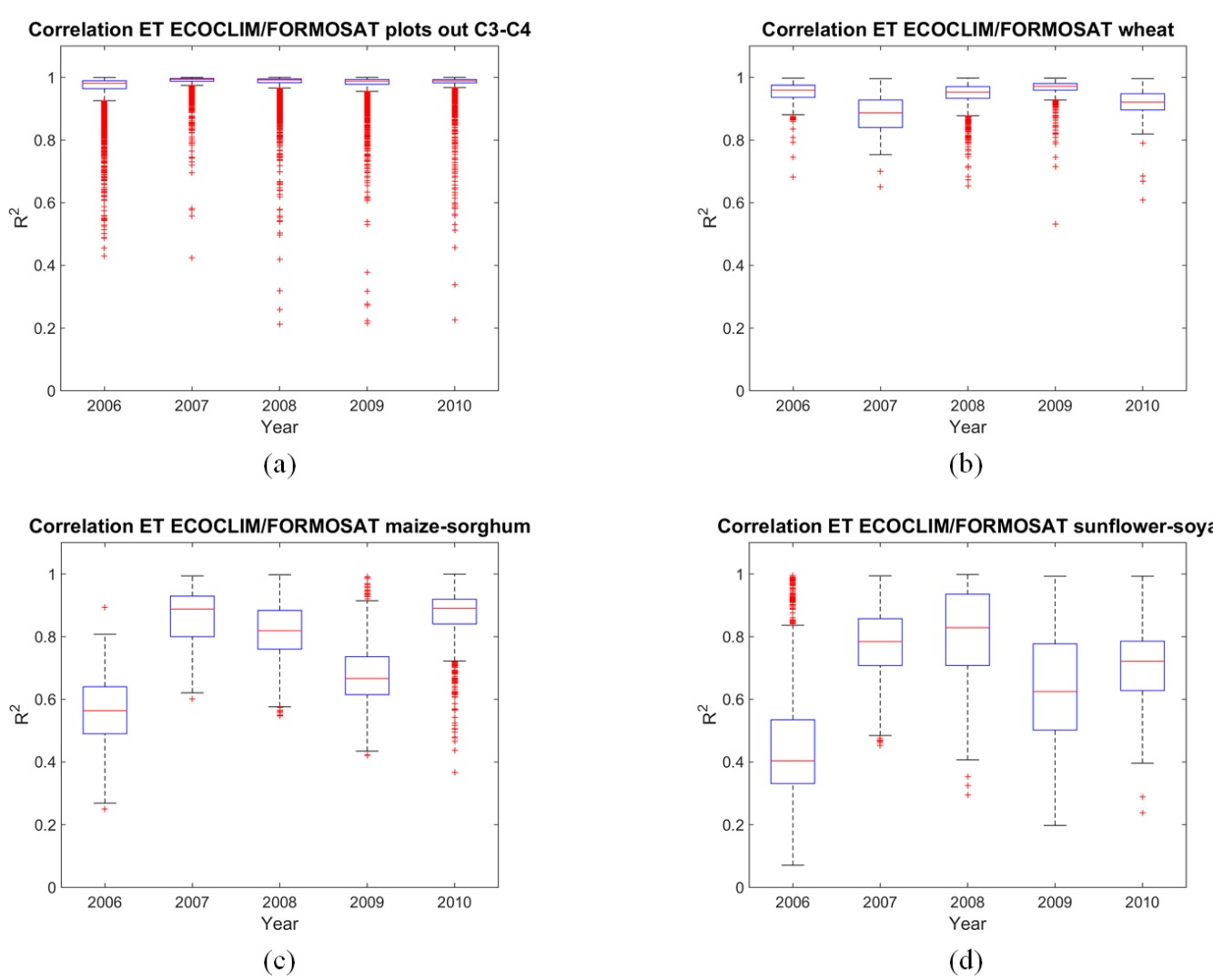

**Figure 7: Correlations between the evapotranspiration time series of the two experiments on: (a) uncultivated plots; (b) wheat crops; (c) C4 crops (maize and sorghum); and (d) sunflower and soy crops. The boxplots show the medians in red. Edges of each box represent quartiles whereas whiskers represent extreme values not considered as outliers (red dots for the outliers).**

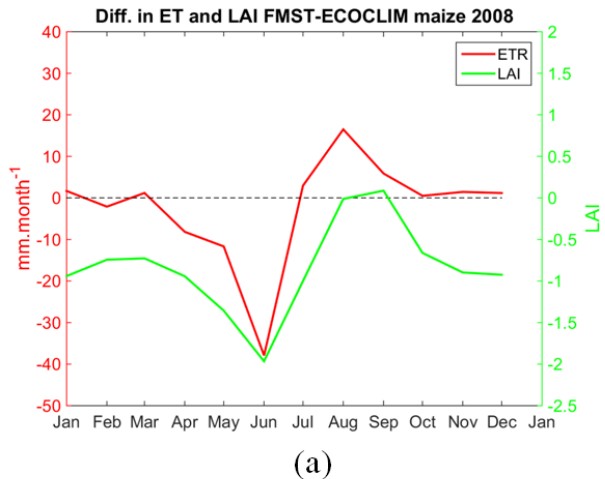

(a)

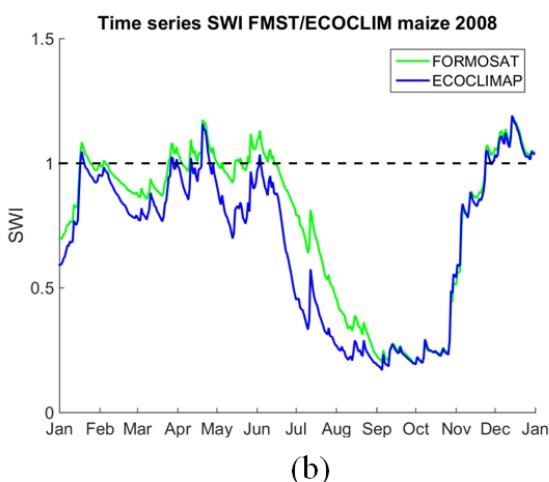

(b)

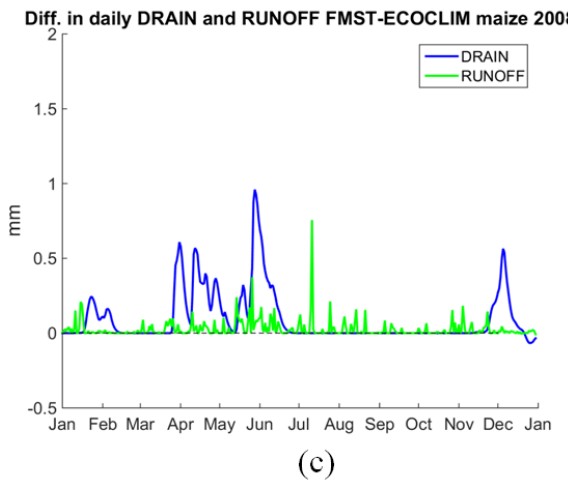

(c)

**Figure 8: (a) Differences (diff.) in LAI (green) and evapotranspiration (red) between the two experiments (FORMOSAT-ECOCLIMAP) ; (b) Time series of Soil Water Index; (c) Differences in drainage (blue) and runoff (green) values; Averaged across C4 crops for 2008**

| Formosat-2 cover map class | SURFEX class |
|---|---|
| 16. Bare soil | 1. Bare soil |
| 4. Urban area<br>33. Gravel pit<br>41. Dense buildings<br>42. Diffuse buildings<br>43. Industrial buildings<br>44. Urban mineral surface | 2. Rocks |
| - | 3. Permanent snow and ice |
| 231. Mixed broadleaf forest<br>2311. Poplar<br>2312. Eucalyptus | 4. Deciduous forest |
| - | 5. Evergreen forest |
| 232. Mixed coniferous forest | 6. Coniferous forest |
| 15. Dual crops<br>121. Wheat<br>122. Barley<br>123. Rapeseed<br>132. Sunflower<br>134. Soya<br>135. Hemp<br>141. Protein plants<br>142. Spring barley<br>1321. Late sunflower<br>1411. Pea | 7. C3 crops |
| 131. Maize<br>133. Sorghum<br>1311. Non-irrigated maize<br>1312. Silage maize | 8. C4 crops |
| - | 9. Irrigated crops |
| 22. & 112. Fallow<br>111. Meadow<br>1111. Temporary meadow<br>1112. Permanent meadow | 10. C3 herbaceous plants |
| - | 11. C4 herbaceous plants |
| 31. River<br>32. Lake | 12. Wetland |

**Table 1: Aggregation rules of Formosat-2 cover maps by SURFEX vegetation type**

| Year | Crop type | R² ECOCLIMAP | R² FORMOSAT | RMSE ECOCLIMAP | RMSE FORMOSAT |
|------|-----------|--------------|-------------|----------------|---------------|
| 2006 | Wheat | 0.90 | 0.94 = | 8.07 | 6.86 = |
| 2007 | Sunflower | 0.66 | 0.89 +++ | 20.9 | 12.1+++ |
| 2008 | Wheat | - | - | - | - |
| 2009 | Rapeseed | 0.97 | 0.96 = | 6.71 | 7.66 = |
| 2010 | Wheat | - | - | - | - |

**Table 2: Correlation coefficient and Root-Mean Square Error of evapotranspiration for the Auradé site**

| Year | Crop type | R² ECOCLIMAP | R² FORMOSAT | RMSE ECOCLIMAP | RMSE FORMOSAT |
|------|-----------|--------------|-------------|----------------|---------------|
| 2006 | Maize | 0.15 | 0.65 +++ | 34.4 | 23.5 +++ |
| 2007 | Wheat | 0.76 | 0.95 ++ | 21.9 | 14.6 ++ |
| 2008 | Maize | 0.58 | 0.82 +++ | 21.5 | 12.4+++ |
| 2009 | Wheat | 0.94 | 0.95 = | 18.7 | 12.6 ++ |
| 2010 | Maize | 0.46 | 0.62 ++ | 27.2 | 22.8 ++ |

5    **Table 3: Correlation coefficient and Root-Mean Square Error of evapotranspiration for the Lamasquère site**

| Vegetation type | 2006 | 2007 | 2008 | 2009 | 2010 | Interannual mean |
|-----------------|------|------|------|------|------|------------------|
| Outside the crops | +15 (+2.9%) | +16 (+2.3%) | +41 (+5.3%) | +20 (+3.1%) | +30 (+4.7%) | +24 (+3.7%) |
| Wheat | -3 (-0.6%) | -1 (-0.2%) | +61 (+8.0%) | +15 (+2.3%) | +20 (+3.1%) | +18 (+2.8%) |
| Sunflower/soya | +5 (+0.9%) | +54 (+7.9%) | +47 (+6.1%) | +30 (+4.7%) | +48 (+7.5%) | +37 (+5.7%) |
| Maize/sorghum | +4 (+0.7%) | +35 (+5.2%) | +35 (+4.6%) | +18 (+2.8%) | +32 (+5.0%) | +25 (+3.8%) |

**Table 4: Differences between FORMOSAT and ECOCLIMAP experiments on the annual drainage level in mm.yr$^{-1}$ and the corresponding fraction of annual precipitations in % (FORMOSAT-ECOCLIMAP).**

| Vegetation type | 2006 | 2007 | 2008 | 2009 | 2010 | Interannual mean |
|---|---|---|---|---|---|---|
| Outside the crops | +3 (+0.6%) | +4 (+0.6%) | +7 (+0.9%) | +4 (+0.6%) | +4 (+0.7%) | **+4 (+0.6%)** |
| Wheat | +1 (+0.2%) | +5 (+0.7%) | +17 (+2.3%) | +7 (+1.1%) | +4 (+0.6%) | **+7 (+1%)** |
| Sunflower/soya | +7 (+1.4%) | +13 (+1.9%) | +10 (+1.4%) | +11 (+1.7%) | +13 (+2.0%) | **+11 (+1.7%)** |
| Maize/sorghum | +6 (+1.1%) | +9 (+1.3%) | +10 (+1.3%) | +9 (+1.4%) | +9 (+1.4%) | **+9 (+1.3%)** |

**Table 5: Differences between FORMOSAT and ECOCLIMAP experiments on annual runoff in mm.yr$^{-1}$ and the corresponding fraction of annual precipitations in % (FORMOSAT-ECOCLIMAP).**

