# Peer review of "Effects of high spatial and temporal resolution Earth observation on simulated hydrometeorological variables in a cropland (southwestern France)"

_Hydrology and Earth System Sciences, 2016_

## Referee Comment (RC1) · Anonymous Referee #1 · 7 Feb 2017

General comment: the objective is significant. the problem of land cover spatial variability and remote sensing estimation at appropriate spatial scale is a key topic. However, several problems and comments are described below and need to be addressed. In particular, I have several doubts on the spatial scales of model, remote sensing observations and eddy covariance fluxes. I think that the paper can be accepted but the following clarifications need to be addressed for properly evaluating the paper.

Specific comments: 1) Introduction: not really clear. You need to write more clearly the objectives and what is the new contribution of the paper. 2) The following are

comments and doubts on spatial scales of remote sensing observations, model and eddy covariance fluxes. What is the height of the eddy covariance tower? What is the foot print length? Are you comparing observed fluxes with modeled fluxes at 1 km resolution? If yes, why? I noted that the foot print of the eddy covariance tower may be not homogenous: are you addressing the spatial variability of the land cover in the foot print? 3) Why are you not running ISBA at finer spatial scales? If you have remote sensing observations at 8 m resolution you can use ISBA at finer spatial scales than 1 km. The use of ISBA at finer spatial scale may help a lot to understand the effect of land cover heterogeneity on land surface fluxes. In this way, you can use properly the remote sensing observations at 8 m spatial resolution. 4) Figure 4: What is the spatial scale? 5) Fig. 5. What is the aggregation scale for comparing LAI values? ECOCLIMAP-II database (1 km resolution) and Formosat-2 database (8 m resolution) are providing different LAI values at the same scale. 6) Figure 5 and 6. You need to show the comparison results for all the simulated period (2006-2010) not just one year. Are the hydrometeorological conditions the same for all the years. Typically Mediterranean regions are characterized by strong interannual variability, hence it is very interesting to evaluate it. in this way you can see the impact of the interannual variability of rainfall seasonality on LAI and fluxes. 7) I'm not sure about figure 7. If you are modeling at 1 km spatial resolution, how can you simulate fluxes of specific cultivations (e.g., wheat, maize-sorghum, etc.)? in a 1 km grid cell you have more than 1 specific cultivation. 8) I'm trying to understand how SURFEX using ECOCLIMAP and SURFEX using FORMOSAT (and GDAL polygonise) are modeling each land cover component. Please, add information and explanations.

---

## Referee Comment (RC2) · Anonymous Referee #2 · 9 Feb 2017

General comments

This paper focuses on the impact of vegetation dynamic on the simulation of evapotranspiration from a land surface model. It shows the benefits of using decametric resolution and high revisit frequency satellite imagery (FORMOSAT-2) to resolve the spatial and temporal dynamic of vegetation at the landscape scale and to drive the SURFEX/ISBA-A-gs land surface model. The authors compare

- evapotranspiration (ET) simulated using the leaf area index (LAI) and a land cover map derived from FORMOSAT-2 satellite imagery,

and

- ET simulated using vegetation variables taken from the ECOCLIMAP-II database which is the land surface parameter database used for the spatial integration of the model and provides a monthly climatology for LAI at 1 km spatial resolution.

The authors showed that the use of FORMOSAT-2 LAI improves the performances of simulated ET. The effects are more significant for summer crops than for winter crops. The issue addressed by this paper is of great interest for the land surface community. It shows the potential of new high spatial and temporal resolution satellite (SENTINEL-2) to drive land surface models using more accurate land surface characteristics.

However, major revisions of the paper are needed before considering it for publication in HESS. There are a lot of confusing sentences, inaccurate definitions, some references are missing, some justifications are missing. The analysis of the results is not deep enough. A dedicated discussion section is missing. This alters the quality of the paper whilst there is enough scientific content for publication. I provide below some evidences and some suggestions for improvement. But this is not exhaustive. Substantial improvement of English and paper structure are also expected.

Specific comments

- Abstract: it is too long, too many methodological details are given

- Introduction - page 2, line 4: please clarify the idea, provide examples of agricultural practices (irrigation, crop rotation, seeding date,...) - choose between Land surface model and SVAT to use in the rest of the text - line 7: references are needed for SURFEX and VIC, - the meaning of SURFEX acronym needs to be given - the definition of LAI is not exact, it is defined as - "half the total developed area of green (i.e., photosynthetic active) leaves per unit ground horizontal surface area [Chen and Black, 1992]" - page2 , line 10: LAI is not an index. It is a variable that can be simulated by the model or used as a forcing variable to drive the model - page 2, line 10: please justify, a reference is needed here. LAI is the scaling factor to compute the stomatal conductance at the canopy scale. It is not necessary the most influential parameter on the simulated evapotranspiration. - Page 2, Line 12: too many references, select one or two. This remark applies for the rest of the paper. - Page 2, Line 14: no needs to defined a climatology - page 2, line 14: use the term climatology instead of climatological - page 20, line 20: This holds for Europe but not for the US. - Page 2, line 20-25: Redundancies, confusing sentences - page 2 line 25-26: why ? References are needed - page 2, line 27-30: redudancies with above - page 2, line 31-32: ISBA should be defined - page 3, line5: ECOCLIMAP-II LAI are derived from the analysis of MODIS LAI and not SPOT/VEGETATION - page 3, line 7-8: this is not clear. Provide thorough explanation on how LAI is computed in ECOCLIMAP-II - page 3, line 10-20: good and clear paragraph

Section 2.1 - not enough model details are given - which version of SURFEX is used ? - Which type of water transfer scheme? Energy balance ? - Is ISBA includes includes a coupled stomatal conductance-photosynthesis scheme (A-gs version) - what about irrigation, is it simulated by the model ïĄň the reference for the ISBA pedotrasnfer function is not correct, use Noilhan, J. and Lacarrère, P.: GCM Grid-Scale Evaporation from Mesocale Modeling, J. Climate, 8, 206–223, 1994 ïĄň -page 4, line 3-5: the description of ECOCLIMAP-II is not accurate. No vegetation parameters are derived from satellite observations. Some parameters are fixed for each plant functional type. Other parameter or variables vary geographically with the type of ecosystem. This part must be properly edited.

Section 2.2: -page 4, line 17: "non-irrigated rotation": this is not a correct term

The authors should provide a dedicated Discussion section. They should properly discuss the main outcomes of the work and discuss their limits. The issue of uncertainties need to be addressed: uncertainty in the measurements, uncertainty in the satellite imagery (registration . . .), uncertainty in the land surface model affecting the simulation of ET ïĄň Conclusion: Conclusion should be re-written giving the main outcomes

of the paper. It should be a summary and should not contain analysis or discussion statements (line 20-25)

English : English must be carefully edited, I provide some examples here

page 2 line 4: "vegetation cover present"→ "present vegetation cover"

page 2 line 5 " the more accurate"→ " more accurate"

page 2 line 6 "to improving"→ "critical to improve"

page 3, line 10 "rather than"→ "instead of "

check in the document the use of "the"

page 3, line 9: vegetation type and LAI

Page 4, line 4-5: "the ISBA"→ "ISBA",

Shorter sentences are needed

the title is too long, some suggestions: use Earth observation instead of remote sensing products use cropland instead of cultivated area use high spatial and temporal resolution

acronyms must be defined

---

## Author Comment (AC1) · 3 Apr 2017

Anonymous Referee #1: "General comment: the objective is significant. The problem of land cover spatial variability and remote sensing estimation at appropriate spatial scale is a key topic. However, several problems and comments are described below and need to be addressed. In particular, I have several doubts on the spatial scales of model, remote sensing observations and eddy covariance fluxes. I think that the paper can be accepted but the following clarifications need to be addressed for properly evaluating the paper. "

Authors: Thank you for your comments, which are pointing out that we should better highlight the issue of the spatial variability of the land cover in current land surface model. Indeed, this issue is the main motivation behind the field scale approach described in the paper. You will find all the explanation about this approach in the following answer to your specific comments. We have done several efforts to make it clearer to the reader. Also, to simplify the comprehension, we have decided to switch the evapotranspiration unit from a monthly averaged J.m-2.d-1 to a cumulated evapotranspiration over the month in mm.month-1. The text, figures and tables have been modified accordingly.

Anonymous Referee #1: "Specific comments: 1) Introduction: not really clear. You need to write more clearly the objectives and what is the new contribution of the paper."

Authors: Efforts have been done to shorten the abstract and clarify the introduction. The objectives are now clearly defined in the new version: "Our study proposes to evaluate the impact of introducing high-resolution information on vegetation type and the LAI from Sentinel-2-like observations rather than the low resolution ECOCLIMAP-II product in ISBA-SURFEX simulations. The main point is to see if the model is able to capture a more accurate phenological cycle, especially the agricultural practices mentioned above, and to simulate its effects on hydrometeorological fluxes ... This way, we will point out the contribution on surface fluxes dynamics of using high spatial and temporal resolution vegetation forcing instead of low resolution climatology" The field scale approach, which constitutes the main novelty of the study, also appears more clearly: "The LSM was applied at the "field" scale to place it under homogeneous vegetation type conditions for each computation unit (only one PFT by unit). This field-scale modeling approach allows one to take into account the spatial variability of LAI values between fields while limiting the computation time in comparison to a pixel-based approach." This approach is described more clearly in the dedicated part (Numerical Experiments, Sect 3.1).

Anonymous Referee #1: "2) The following are comments and doubts on spatial scales

of remote sensing observations, model and eddy covariance fluxes. What is the height of the eddy covariance tower? What is the foot print length? Are you comparing observed fluxes with modeled fluxes at 1 km resolution? If yes, why? I noted that the foot print of the eddy covariance tower may be not homogenous: are you addressing the spatial variability of the land cover in the foot print?"

Authors: The eddy covariance tower is 3.65m high on Lamasquère and 2.85m high on Auradé. A sentence has been added in the manuscript about it: "Each flux site is equipped with 1) eddy covariance systems to measure half-hourly sensible heat flux and evapotranspiration, installed at 2.8 and 3.65 meters above the soil at Auradé and Lamasquère, respectively" The tower location and data filtering insures that the footprint is totally included in the field when data are available, in accordance with the Carbo-Europe and GAG-Europe experimental protocols. Thus the vegetation in the footprint is homogeneous. A paragraph has been added to the section 2.2.3 to explain the filtering criteria: "These scalars are measured at 20 Hz and are integrated over 30 minutes to generate surface fluxes according to CarboEurope-IP flux computation and filtering procedures (Aubinet & al., 1999, Béziat & al, 2009). Thus, flux data (NEE, LE and H) were filtered to remove outliers and out of range data. We applied the recommended filtering criteria concerning 1) periods of low turbulence and tests on stationarity (Papale et al., 2006; Reichstein et al., 2005) because the Eddy Covariance assumptions are not fulfilled in those cases; 2) periods of rain because they disturb the signal of both the open path analyser, and the sonic anemometer, and 3) eventually, the spatial representativeness (footprint) of the fluxes. For this third filter, a fetch including 90 % of the flux was computed with the Kljun & al. (2004) model for each half-hourly EC flux value (F-90).Then, this fetch was compared with the distance between the mast and the edge of the plot in the main wind direction (D). If F-90>D, fluxes were discarded because we assumed that it was not sufficiently representative of the plot." These measured fluxes were compared to the modeled fluxes at the field scale. This field scale approach for the simulations is described in the answers below. It is also more accurately described in the new version of the paper.

Anonymous Referee #1: "3) Why are you not running ISBA at finer spatial scales? If you have remote sensing observations at 8 m resolution you can use ISBA at finer spatial scales than 1 km. The use of ISBA at finer spatial scale may help a lot to understand the effect of land cover heterogeneity on land surface fluxes. In this way, you can use properly the remote sensing observations at 8 m spatial resolution"

Authors: As presented in the abstract, the introduction and the section 3.1, we used a field scale approach for both our experiments. This approach constitutes the novelty of the study. It consists in doing simulations on an irregular grid where each calculation cell is a parcel, geolocalized by its centroid, defined by a polygon and associated to homogeneous vegetation (PFT). These plots are the ones determined from the Formosat-2 land cover maps (with GDAL_polygonize). The field scale seemed to us like a pertinent working scale for two reasons: - It is a coherent functional landscape unit with homogeneous vegetation dynamic and thus hydro-meteorological behavior. - It allows exploiting the high spatial resolution of Formosat-2 while limiting the calculation time compared to a pixel based approach at the resolution of Formosat. Running at 8m resolution is beyond the scope of the study and computationally intractable for such a large area. Efforts have been done throughout the entire manuscript to make this point clearer to the reader.

Anonymous Referee #1: "4) Figure 4: What is the spatial scale?"

Authors: This figure represents the study area which is a square with a 24km side. The scale has been added to the figure (cf. figure at the end)

Anonymous Referee #1: "5) Fig. 5. What is the aggregation scale for comparing LAI values? ECOCLIMAP-II database (1 km resolution) and Formosat-2 database (8 m resolution) are providing different LAI values at the same scale."

Authors: As described in the answer to your point 3), both our experiments were done at the field scale. The Formosat-2 LAI was calculated by averaging the pixel values in each plot. An erosion was applied to the plots, with a 16m value (twice the size of

the Formosat-2 resolution) to avoid border effects and geo-location uncertainties of the remote sensing product. The method of LAI retrieval is now more accurately described in the section 2.2.1. Each ECOCLIMAP-II grid cell is described by a composition of vegetation types (patches, Sect. 2.1). Each patch has its own LAI cycle derived from MODIS products (Faroux et al. 2013). In the reference simulation, the patches and corresponding LAI values for each field are taken from the nearest ECOCLIMAP-II regular grid cell (with a 1km resolution). Hence the comparison between the two experiments is done on each field by focusing on the same patch (i.e. the one given by the Formosat-2 land cover map). A sentence has been added in the section 3.2 to explain this: "Each plot has a unique patch in the FORMOSAT experiment, forced by the land cover map. Thus only the corresponding patch was taken into account when comparing with the ECOCLIMAP experiment. In the spatial comparison, if the corresponding patch was not present in the combination of patches of the plot, then this plot is excluded of the results. By this way we are sure that we can compare the fluxes on specific vegetation types. "

Anonymous Referee #1: "6) Figure 5 and 6. You need to show the comparison results for all the simulated period (2006-2010) not just one year. Are the hydrometeorological conditions the same for all the years. Typically Mediterranean regions are characterized by strong interannual variability, hence it is very interesting to evaluate it. in this way you can see the impact of the interannual variability of rainfall seasonality on LAI and fluxes."

Authors: The figures 5 and 6 are only meant to support a discussion on the ongoing processes affected. Of course the comparison has been done over the whole period and leads to the same conclusions. A year-to-year variability is visible due to the changes in agricultural practices, which are closely related to the climatic conditions of the year. The tables 2 and 3 summarize these results by showing the correlation coefficient and the root mean square error between each experiment and in-situ measurements for both sites. It points out a systematic enhancement of the scores with

amplitude depending of the year. Indeed, if the ECOCLIMAP-II LAI dynamic is closer to the measured and remotely sensed ones, the improvement is weaker and inversely. A sentence about this issue has been added in a new discussion section (Sect. 5 in the new manuscript): "The dependence on the year for the results on evapotranspiration (Fig. 7) may be justified by the climatic conditions of each year. Indeed, climatic conditions influence the farmers' decision concerning the seeding and/or harvest dates. If these dates are closer than the ones simulated by ECOCLIMAP LAI, the effect on evapotranspiration is weaker."

Anonymous Referee #1: "7) I'm not sure about figure 7. If you are modeling at 1 km spatial resolution, how can you simulate fluxes of specific cultivations (e.g., wheat, maize-sorghum, etc.)? in a 1 km grid cell you have more than 1 specific cultivation."

Authors: As said in the previous answers, we have done the simulations at the field scale. In the reference simulation, each field is represented by a combination of the 12 patches available in SURFEX. ISBA simulates the fluxes on each patch separately so you can choose each of these patches when you interpret the results. To compare on a specific cultivation, you just have to choose the corresponding patch in the results of the simulation. As describe in our answer to your point 5), a sentence has been added to clarify this point.

Anonymous Referee #1: "8) I'm trying to understand how SURFEX using ECOCLIMAP and SURFEX using FORMOSAT (and GDAL polygonise) are modeling each land cover component. Please, add information and explanations. "

Authors: The answers to your previous comments may have given you the answer to this one. In the reference simulation, the forcing of ECOCLIMAP-II is taken from the nearest regular grid cell for each vegetation type (patch). All these patches are simulated separately by SURFEX so you can focus on a specific patch for the results. For plots belonging to the same ECOCLIMAP-II grid cell, only parameters given by another set of forcing data than ECOLIMAP-II may change. It may be the case of the soil parameters or the meteorological forcing if they are not superposed to ECOCLIMAP-II. Also, the initialization of the soil temperature and water content may not be the same for all these plots. Indeed, our simulation grid changes every year as the land cover map changes too. Thus the plots are not exactly the same from year to year due to the polygonal segmentation with GDAL. To initialize the soil temperature and water content for each plot and each year, we use an interpolation using the inverse distance method on the 9 nearest neighboring plots in the previous year grid. To initialize the first year of simulation, we have done a simulation on the same grid but using the meteorological forcing of the year before. Your comments let us think that our manuscript was probably not clear enough regarding the use of the crop field as a computation unit. We hope that our explanations and the modifications made to manuscript have clarified this point in particular, despite the relative complexity of the unusual way we use SURFEX-ISBA. We think that your comments helped improve the clarity of the paper. We thank you again sincerely for your evaluation of our work.
* * *
Fig. 1.

| | |
|---|---|
| | Unclassed |
| | Broadleaf forest |
| | Coniferous forest |
| | Eucalyptus |
| | Wheat |
| | Rapeseed |
| | Barley |
| | Maize |
| | Sunflower |
| | Sorghum |
| | Soya |
| | Pea |
| | Fallow |
| | Fallow |
| | Temporary meadow |
| | River |
| | Lake |
| | Dense buildings |
| | Industrial buildings |
| | Gravel pit |
| | Poplar |
| | Permanent meadow |
| | Diffuse buildings |
| | Urban mineral surface |
| | Silage maize |

---

## Author Response (AR1)

Dear Editor and Referees,

We would like to thank you for your helpful comments. We have done many modifications to the manuscript according to your suggestions. Please find below the point-by-point reply to the referees comments and a marked-up version of the manuscript.

We also wish to bring the Editor's attention to the fact that we would like to add Dr Tiphaine Tallec as a co-author of the paper. Indeed, her contribution was instrumental in addressing the issue of the eddy covariance measurements uncertainties that was raised by Reviewer 2.

Sincerely yours.

The authors

**Anonymous Referee #1:** *"General comment: the objective is significant. The problem of land cover spatial variability and remote sensing estimation at appropriate spatial scale is a key topic. However, several problems and comments are described below and need to be addressed. In particular, I have several doubts on the spatial scales of model, remote sensing observations and eddy covariance fluxes. I think that the paper can be accepted but the following clarifications need to be addressed for properly evaluating the paper. "*

Authors: Thank you for your comments, which are pointing out that we should better highlight the issue of the spatial variability of the land cover in current land surface model. Indeed, this issue is the main motivation behind the plot scale approach described in the paper. You will find all the explanation about this approach in the following answer to your specific comments. We have done several efforts to make it clearer to the reader. Also, to simplify the comprehension, we have decided to switch the evapotranspiration unit from a monthly averaged $J.m^{-2}.d^{-1}$ to a cumulated evapotranspiration over the month in $mm.month^{-1}$. The text, figures and tables have been modified accordingly.

**Anonymous Referee #1:** *"Specific comments: 1) Introduction: not really clear. You need to write more clearly the objectives and what is the new contribution of the paper."*

Authors: Efforts have been done to shorten the abstract and clarify the introduction. The objectives are now clearly defined in the new version:

> *Our study aims at evaluating the impact of introducing high-resolution information on vegetation type and LAI from Sentinel-2-like observations instead of the low resolution climatology of ECOCLIMAP-II in ISBA-SURFEX simulations. The main objective is to assess if this more accurate description of the phenological cycle, especially the agricultural practices mentioned above, translates into a better representation of the simulated evapotranspiration. We also aim at evaluating the impact on simulated drainage and runoff.*
> *…*
> *This way, we will point out the contribution on surface fluxes dynamics of using high spatial and temporal resolution vegetation forcing instead of a low resolution climatology.*

The plot scale approach, which constitutes the main novelty of the study, also appears more clearly:

> *The LSM was applied at the "plot" scale to place it under homogeneous vegetation type conditions for each computation unit (one computation unit has only one PFT). This plot-scale modeling approach allows us to take into account the spatial variability of LAI values between plots while limiting the computation time in comparison to a pixel-based approach.*

This approach is described more clearly in the dedicated part (Numerical Experiments, Sect 3.1).

**Anonymous Referee #1:** *"2) The following are comments and doubts on spatial scales of remote sensing observations, model and eddy covariance fluxes. What is the height of the eddy covariance tower? What is the foot print length? Are you comparing observed fluxes with modeled fluxes at 1 km resolution? If yes, why? I noted that the foot print of the eddy covariance tower may be not homogenous: are you addressing the spatial variability of the land cover in the foot print?"*

Authors: The eddy covariance tower is 3.65m high on Lamasquère and 2.8m high on Auradé. A sentence has been added in the manuscript about it:

*Each flux site is equipped with 1) eddy covariance systems to measure half-hourly sensible heat flux and evapotranspiration, installed at 2.8 and 3.65 meters above the soil at Auradé and Lamasquère sites, respectively*

The tower location and data filtering insures that the footprint is totally included in the field when data are available, in accordance with the Carbo-Europe and GAG-Europe experimental protocols. Thus the vegetation in the footprint is homogeneous.

A paragraph has been added to the section 2.2.3 to explain the filtering criteria:

*Half-hourly fluxes were corrected for spectral frequency loss (Moore, 1986) and corrected for air density variations (Webb et al., 1980). Flux data were filtered and flagged according to statistics and objectives criteria: data out of range, rain event, friction velocity threshold, integral turbulence characteristic, stationarity test (Papale et al., 2006; Reichstein et al., 2005) and spatial representativeness (footprint) of the fluxes. For the latter, if the calculated fetch including 90 % of the flux (Kljun et al., 2004) model for each half-hourly EC flux value (F-90) was higher than the distance between the mast and the edge of the plot in the main wind direction, fluxes were discarded. Gapfilling was finally performed depending on the duration of missing data, either following the linear regression method (duration < 1h30), or following the mean diurnal variation or look up table method (duration >1h30) according to Beziat & al. (2009).*

These measured fluxes were compared to the modeled fluxes at the plot scale. This plot scale approach for the simulations is described in the answers below. It is also more accurately described in the new version of the paper.

**Anonymous Referee #1:** *"3) Why are you not running ISBA at finer spatial scales? If you have remote sensing observations at 8 m resolution you can use ISBA at finer spatial scales than 1 km. The use of ISBA at finer spatial scale may help a lot to understand the effect of land cover heterogeneity on land surface fluxes. In this way, you can use properly the remote sensing observations at 8 m spatial resolution"*

Authors: As presented in the abstract, the introduction and the section 3.1, we used a plot scale approach for both our experiments. This approach constitutes the novelty of the study. It consists in doing simulations on an irregular grid where each calculation cell is a plot, geolocalized by its centroid, defined by a polygon and associated to homogeneous vegetation (PFT). These plots are the ones determined from the Formosat-2 land cover maps (with GDAL_polygonize).

The plot scale seemed to us like a pertinent working scale for two reasons:

- It is a coherent functional landscape unit with homogeneous vegetation dynamic and thus hydro-meteorological behavior.
- It allows exploiting the high spatial resolution of Formosat-2 while limiting the calculation time compared to a pixel based approach at the resolution of Formosat. Running at 8m resolution is beyond the scope of the study and computationally intractable for such a large area.

Efforts have been done throughout the entire manuscript to make this point clearer to the reader.

**Anonymous Referee #1:** *"4) Figure 4: What is the spatial scale?"*

Authors: This figure represents the study area which is a square with a 24km side. The scale has been added to the figure.

[Figure]

**Anonymous Referee #1:** *"5) Fig. 5. What is the aggregation scale for comparing LAI values? ECOCLIMAP-II database (1 km resolution) and Formosat-2 database (8 m resolution) are providing different LAI values at the same scale."*

Authors: As described in the answer to your point 3), both our experiments were done at the field scale. The Formosat-2 LAI was calculated by averaging the pixel values in each plot. An erosion was applied to the plots, with a 16m value (twice the size of the Formosat-2 resolution) to avoid border effects and geo-location uncertainties of the remote sensing product. The method of LAI retrieval is now more accurately described in the section 2.2.1.

Each ECOCLIMAP-II grid cell is described by a composition of vegetation types (patches, Sect. 2.1). Each patch has its own LAI cycle derived from MODIS products (Faroux et al. 2013). In the reference simulation, the patches and corresponding LAI values for each field are taken from the nearest ECOCLIMAP-II regular grid cell (with a 1km resolution). Hence the comparison between the two experiments is done on each field by focusing on the same patch (i.e. the one given by the Formosat-2 land cover map).

A sentence has been added in the section 3.2 to explain this:

*Each plot has a unique ISBA patch in the FORMOSAT experiment, forced by the land cover map. Thus only the corresponding patch was taken into account when comparing with the ECOCLIMAP experiment. If the corresponding patch was not present in the combination of patches given by ECOCLIMAP-II for the plot, then this plot was excluded of the results. By this way we are sure that we can compare the fluxes on specific vegetation types.*

**Anonymous Referee #1:** *"6) Figure 5 and 6. You need to show the comparison results for all the simulated period (2006-2010) not just one year. Are the hydrometeorological conditions the same for all the years. Typically Mediterranean regions are characterized by strong interannual variability, hence it is very interesting to evaluate it. in this way you can see the impact of the interannual variability of rainfall seasonality on LAI and fluxes."*

Authors: The figures 5 and 6 are only meant to support a discussion on the ongoing processes affected. Of course the comparison has been done over the whole period and leads to the same conclusions. A year-to-year variability is visible due to the changes in agricultural practices, which are closely related to the climatic conditions of the year. The tables 2 and 3 summarize these results by showing the correlation coefficient and the root mean square error between each experiment and in-situ measurements for both sites. It points out a systematic enhancement of the scores with amplitude depending of the year. Indeed, if the ECOCLIMAP-II LAI dynamic is closer to the measured and remotely sensed ones, the improvement is weaker and inversely. A sentence about this issue has been added in a new discussion section (Sect. 5 in the new manuscript):

> *The interannual variability of the results on evapotranspiration (Fig. 7) may be justified by the climatic conditions of each year. Indeed, climatic conditions influence the farmers' decisions concerning the seeding and/or harvest dates. If these dates are closer than the ones simulated by ECOCLIMAP LAI, the effect on evapotranspiration is weaker.*

**Anonymous Referee #1:** *"7) I'm not sure about figure 7. If you are modeling at 1 km spatial resolution, how can you simulate fluxes of specific cultivations (e.g., wheat, maize-sorghum, etc.)? in a 1 km grid cell you have more than 1 specific cultivation."*

Authors: As said in the previous answers, we have done the simulations at the field scale. In the reference simulation, each field is represented by a combination of the 12 patches available in SURFEX. ISBA simulates the fluxes on each patch separately so you can choose each of these patches when you interpret the results. To compare on a specific cultivation, you just have to choose the corresponding patch in the results of the simulation. As describe in our answer to your point 5), a sentence has been added to clarify this point.

**Anonymous Referee #1:** *"8) I'm trying to understand how SURFEX using ECOCLIMAP and SURFEX using FORMOSAT (and GDAL polygonise) are modeling each land cover component. Please, add information and explanations. "*

Authors: The answers to your previous comments may have given you the answer to this one. In the reference simulation, the forcing of ECOCLIMAP-II is taken from the nearest regular grid cell for each vegetation type (patch). All these patches are simulated separately by SURFEX so you can focus on a specific patch for the results. For plots belonging to the same ECOCLIMAP-II grid cell, only parameters given by another set of forcing data than ECOLIMAP-II may change. It may be the case of the soil parameters or the meteorological forcing if they are not superposed to ECOCLIMAP-II. Also, the initialization of the soil temperature and water content may not be the same for all these plots. Indeed, our simulation grid changes every year as the land cover map changes too. Thus the plots are not exactly the same from year to year due to the polygonal segmentation with GDAL. To initialize the soil temperature and water content for each plot and each year, we use an interpolation using the inverse distance method on the 9 nearest neighboring plots in the previous year grid. To initialize

the first year of simulation, we have done a simulation on the same grid but using the meteorological forcing of the year before.

Your comments let us think that our manuscript was probably not clear enough regarding the use of the crop field as a computation unit. We hope that our explanations and the modifications made to manuscript have clarified this point in particular, despite the relative complexity of the unusual way we use SURFEX-ISBA. We think that your comments helped improve the clarity of the paper. We thank you again sincerely for your evaluation of our work.

**Anonymous Referee #2:** *"This paper focuses on the impact of vegetation dynamic on the simulation of evapotranspiration from a land surface model. It shows the benefits of using decametric resolution and high revisit frequency satellite imagery (FORMOSAT-2) to resolve the spatial and temporal dynamic of vegetation at the landscape scale and to drive the SURFEX/ISBA-A-gs land surface model. The authors compare*
*- evapotranspiration (ET) simulated using the leaf area index (LAI) and a land cover map derived from FORMOSAT-2 satellite imagery, and*
*- ET simulated using vegetation variables taken from the ECOCLIMAP-II database which is the land surface parameter database used for the spatial integration of the model and provides a monthly climatology for LAI at 1 km spatial resolution.*

*The authors showed that the use of FORMOSAT-2 LAI improves the performances of simulated ET. The effects are more significant for summer crops than for winter crops. The issue addressed by this paper is of great interest for the land surface community. It shows the potential of new high spatial and temporal resolution satellite (SENTINEL-2) to drive land surface models using more accurate land surface characteristics. However, major revisions of the paper are needed before considering it for publication in HESS. There are a lot of confusing sentences, inaccurate definitions, some references are missing, some justifications are missing. The analysis of the results is not deep enough. A dedicated discussion section is missing. This alters the quality of the paper whilst there is enough scientific content for publication. I provide below some evidences and some suggestions for improvement. But this is not exhaustive. Substantial improvement of English and paper structure are also expected."*

Authors: Thank you for your comments. Several efforts have been done to clarify the paper. The references and justifications missing have been added. We have also decided to switch the evapotranspiration unit from a monthly averaged $J.m^{-2}.d^{-1}$ to a cumulated evapotranspiration over the month in $mm.month^{-1}$ in order to simplify the comprehension. Concerning the analysis of the results, as recommended, we have added a dedicated discussion part where we interpret each interesting aspects of the results. Especially, further work has been done to include a reflection about the uncertainties of both the measurements and the remote sensing products.

Regarding the English improvement, we would like to mention that the paper was revised by American Journal Experts before the submission. The certificate can be downloaded from the Editor portal here: [https://secure.aje.com/download.php?action=certificate&key=BCD9-C850-DE8C-8F39-066A&_t1490784542](https://secure.aje.com/download.php?action=certificate&key=BCD9-C850-DE8C-8F39-066A&_t1490784542) . However, we did our best to further improve the English in the revised version.

**Anonymous Referee #2:** *"Specific comments*

*- Abstract: it is too long, too many methodological details are given"*

Authors: The abstract has been shortened. Methodological details have been suppressed while the main issues and outcomes of the study have been highlighted.

**Anonymous Referee #2:** *"Introduction - page 2, line 4: please clarify the idea, provide examples of agricultural practices (irrigation, crop rotation, seeding date,. . .)*

Authors: This sentence was clarified as follows:

> *In an agricultural river basin, farmer's practices have an impact on crop functioning. Farmers manage crop rotations, select variety, decide the seeding and harvest dates and organize irrigation supplements. In such basins, a more accurate description of crop dynamics and their effects on hydrometeorological fluxes is critical to improve the monitoring of water resources (Foley et al., 2005; Martin et al. 2016).*

**Anonymous Referee #2:** *"- choose between Land surface model and SVAT to use in the rest of the text"*

Authors: We decided to choose "Land Surface Model" for the entire article.

**Anonymous Referee #2:** *"line 7: references are needed for SURFEX and VIC, - the meaning of SURFEX acronym needs to be given"*

Authors: Because SURFEX is not exactly a scientific model but rather a modeling platform, we decided to use ISBA in this sentence. The ISBA acronym is defined and the references are added for both VIC and ISBA:

> *Land surface models (LSMs), such as the Variable Infiltration Capacity (VIC, Liang & al., 1994) or Interactions between the Surface Biosphere Atmosphere (ISBA, Noilhan & Planton, 1989) models …*

**Anonymous Referee #2:** *"- the definition of LAI is not exact, it is defined as - "half the total developed area of green (i.e., photosynthetic active) leaves per unit ground horizontal surface area [Chen and Black, 1992]"*

Authors: Thank you for this comment. This definition and the associated reference have been added in the paper.

**Anonymous Referee #2:** *"page2 , line 10: LAI is not an index. It is a variable that can be simulated by the model or used as a forcing variable to drive the model - page 2, line 10: please justify, a reference is needed here. LAI is the scaling factor to compute the stomatal conductance at the canopy scale. It is not necessary the most influential parameter on the simulated evapotranspiration"*

Authors: As you wrote, the LAI is used to compute the stomatal conductance. It is not necessarily the most influential parameter on the simulated evapotranspiration. The meaning of this sentence was to say that the Leaf Area Index is the only variable representative of the vegetation dynamic to impact evapotranspiration calculation in most LSMs. As described in Noilhan & Planton (1989), the LAI is the only way the vegetation dynamic is taken into account in the evapotranspiration computation in the standard version of ISBA. The root-depth also impacts the maximum available water content to evapotranspiration but in our study, it is a fixed parameter given by ECOCLIMAP. The LAI is also the only vegetation variable to impact the evapotranspiration in VIC (Liang & al., 1994), in the Canadian Land Surface Scheme (CLASS, Verseghy & al., 1993) and the Joint UK Land Environment Simulator (JULES, Best & al., 2011). The other variables that influence the vegetative part of the evapotranspiration are either atmospheric or soil parameters.

The advantage of focusing on the LAI is also that it is a biophysical variable which is observable from space.

The sentence has thus been revised:

> *It is the main variable used to parameterize the effect of vegetation dynamics on evapotranspiration in most LSMs.*

**Anonymous Referee #2: "*Page 2, Line 12: too many references, select one or two. This remark applies for the rest of the paper.*"**

Authors: We kept Verseghy & al. (1933) and Maurer & al. (2002).

**Anonymous Referee #2: "*Page 2, Line 14: no needs to define a climatology - page 2, line 14: use the term climatology instead of climatological*"**

Authors: The term "climatological" has been replaced by "climatology" in the entire paper.

**Anonymous Referee #2: "*page 2, line 20: This holds for Europe but not for the US*"**

Authors: We totally agree with this remark. Indeed, MODIS could be sufficient in North America. This is probably the reason why Sentinel-2 is an EU program. Fritz & al. (2015) have done a global map of field size which could justify this sentence.

The precision has been added:

> *However, for many cultivated areas and particularly in European countries (Fritz & al., 2015), field plot areas rarely exceed typical MODIS product pixel sizes (500 m, i.e., 25 hectares).*

**Anonymous Referee #2: "*Page 2, line 20-25: Redundancies, confusing sentences*"**

Authors: This part has been revised to suppress the redundancies and try to make the idea clearer to the reader:

> *As a result, MODIS pixels can contain mixed LAI signatures of different crop types with different phenologies. It thus degrades the actual temporal variability of the LAI on these fields. Consequently, it is not representative of the actual hydrometeorological behavior of each land cover type (Trezza et al., 2013; Nagler et al., 2013).*

**Anonymous Referee #2: "*page 2 line 25-26: why ? References are needed*"**

Authors: Summer and winter crops have anti-correlated phenologies. So if their LAI signatures are mixed, the resulting cycle will be a nearly constant phenology throughout the year. The variability of the LAI will be attenuated or even suppressed. A sentence has been added to explain this:

> *Indeed, summer and winter crops have anti-correlated phenologies so mixing these two LAI signatures leads to attenuating, or even suppressing, the LAI variability throughout the year.*

However, there are no references in our knowledge that show this phenomenon.

**Anonymous Referee #2: "*page 2, line 27-30: redundancies with above*"**

Authors: These sentences have been reformulated to suppress the redundancies:

> *A potential solution to access realistic vegetation dynamic could be the use of high resolution remote sensing products. The recently launched Sentinel-2 mission generates multispectral imagery of land areas at a decametric resolution (10 m to 60 m depending on the band) over a 5-day revisit period with global coverage. Previous studies have already shown that higher resolution data can improve descriptions of vegetation and modeled water processes in agricultural landscapes for which mid-resolution imagery is unsuitable (Ferrant & al., 2014; Ferrant & al., 2016)*

We found it important to show that some studies have already been carried out to evaluate the impact of the high resolution remote sensing products.

**Anonymous Referee #2:** *"page 2, line 31-32: ISBA should be defined"*

Authors: The ISBA acronym is defined and referenced (Noilhan & Planton, 1989).

**Anonymous Referee #2:** *"page 3, line5: ECOCLIMAP-II LAI are derived from the analysis of MODIS LAI and not SPOT/VEGETATION"*

Authors: You are perfectly right, it has been corrected.

> *It is a climatology derived from MODIS satellite observations collected between 1999 and 2005.*

**Anonymous Referee #2:** *"page 3, line 7-8: this is not clear. Provide thorough explanation on how LAI is computed in ECOCLIMAP-II"*

Authors: The ECOCLIMAP-II is produced by the mean of an unmixing algorithm applied to MODIS LAI product. For each MODIS pixel, ECOCLIMAP-II gives a combination of vegetation types (patches or PFT) with their corresponding fractions. The algorithm takes the nearest MODIS pixel with a pure vegetation type to distinguish the contribution of each patch in the mixed pixel signature. The detail of the method is given by Faroux & al. (2013). This method is briefly described in the introduction and in the presentation of the model (Sect. 2.1) within the new version of the manuscript:

> *Because of the low spatial resolution of MODIS, the LAI signatures of several vegetation types are often mixed in a pixel. An unmixing method is then used by Faroux et al. (2013). It uses the nearest unmixed pixels of each PFT present in the MODIS pixel considered to assess the contribution of each PFT in the LAI climatology.*

**Anonymous Referee #2:** *"Section 2.1 - not enough model details are given - which version of SURFEX is used ? … Is ISBA includes includes a coupled stomatal conductance-photosynthesis scheme (A-gs version)"*

Authors: We used SURFEX 7.3. ISBA is used in its standard version and not the A-gs one. The A-gs version has been tested with the AST option which includes two strategies of plant's response to water stress. It led to the same conclusions with few modifications of the monthly fluxes.
These precisions have been added to the description of the model (Sect. 2.1) :

> *In this study, we used the Interactions between Surface Biosphere Atmosphere (ISBA, Noilhan et Planton, 1989) nature model included in the version 7-3 of SURFEX. The ISBA model uses meteorological and physiographic data to simulate energy and water fluxes between land surfaces and the atmosphere (Fig. 1).*

> *The version used in this study is the standard version of ISBA. It does not include a coupled stomatal conductance-photosynthesis scheme like in the A-gs version (Calvet et al., 1998).*

**Anonymous Referee #2:** *"Which type of water transfer scheme? Energy balance ?"*

Authors: For the water transfer scheme, we used the force restore approach (Deardorff, 1977, included in ISBA by Mahfouf & Noilhan, 1996) with three soil layers (Boone & al., 1999). For the runoff calculation, we used the Variable Infiltration Capacity approach (Dumenil & Todini, 1992) to calculate a subgrid runoff even if the soil layers of the cell are not saturated. This approach has been included in ISBA by Habets & al., 1999. The energy balance scheme is a single source scheme, i.e. computing a unique surface temperature for the soil and the vegetation. It uses the concept of stomatal resistance introduced by Jarvis (1976).

These precisions have been added in the section 2.1 and the new references with it :

> *The water transfer in the soil is simulated on three layers with a force-restore approach presented by Deardorff (1977). This approach was integrated in ISBA by Mahfouf et Noilhan (1996). The three layers were described and calibrated by Boone et al. (1999). The surface layer volumetric water content is restored depending on the water content of both surface and root zone layer. A gravitational drainage flux is simulated when the soil water content of a layer exceeds the field capacity. In the version we used, a subgrid runoff is also calculated using the Variable Infiltration Capacity scheme first described by Dumenil and Todini (1992) and included in SURFEX by Habets et al. (1999). It allows simulating a runoff flux even when the soil is not fully saturated. A unique energy budget is simulated on the vegetation-soil layers composite by a single source scheme. A single surface temperature is used to compute the different energy fluxes. The method is detailed by Noihlan and Planton (1989).*

**Anonymous Referee #2:** *"what about irrigation, is it simulated by the model"*

Authors: As said in the results and conclusion, the irrigation is not simulated by the model in this study. The first reason is that we do not have access to spatialized forcing of irrigation yet, neither in volume nor on the determination of irrigation area. The second reason is that without taking irrigation into account, we can isolate the effect of the LAI modification from the effect of adding irrigation. The third reason is also that there is no automatic module of irrigation for this version of ISBA yet. Currently, ISBA can simulate an automatic irrigation pattern only in the A-gs versions with interactive vegetation, i.e. with a simulated LAI. The irrigation issue is one of the main outcomes of this study and the manuscript has been modified to highlight it. This key point will be the main focus of our future work.

**Anonymous Referee #2:** *"the reference for the ISBA pedotrasnfer function is not correct, use Noilhan, J. and Lacarrère, P.: GCM Grid-Scale Evaporation from Mesocale Modeling, J. Climate, 8, 206–223, 1994"*

Authors: We have replaced the previous reference (Masson & al., 2013) by the one you proposed.

**Anonymous Referee #2:** *"page 4, line 3-5: the description of ECOCLIMAP-II is not accurate. No vegetation parameters are derived from satellite observations. Some parameters are fixed for each plant functional type. Other parameter or variables vary geographically with the type of ecosystem. This part must be properly edited.*

Authors: As described by Faroux & al. (2013), the ecosystems (or cover) are deduced from a classification based on SPOT/VEGETATION Normalized Difference Vegetation Index, crossed with already existing land cover, soil and climate maps. Each LAI profile of each vegetation type of each

cover is then deduced from the MODIS LAI product as briefly described above. This is the only things that are determined by satellite observations. The other parameters come from different sources as described in Masson & al. (2003). This part has been properly modified to be more accurate.

**Anonymous Referee #2:** *"Section 2.2: -page 4, line 17: "non-irrigated rotation": this is not a correct term"*

Authors: The term has been rectified:

> *The two main types of crops found in this area are irrigated summer crops such as maize or soy plants and rain-fed rotation crops such as wheat and sunflower plants.*

**Anonymous Referee #2:** *"The authors should provide a dedicated Discussion section. They should properly discuss the main outcomes of the work and discuss their limits. The issue of uncertainties need to be addressed: uncertainty in the measurements, uncertainty in the satellite imagery (registration . . .), uncertainty in the land surface model affecting the simulation of ET"*

Authors: The results and conclusion has been revised to provide a dedicated discussion part, as you suggested. We discuss the fact that the Formosat-2 LAI allows distinguishing the actual phenologic cycle and particularly the agricultural practices that modifies this cycle like seeding, harvest or crop rotations. The impact on evapotranspiration is then analyzed, showing the limits of the unmixing algorithm of ECOCLIMAP-II LAI retrieval method. It also shows the issue of the lack of irrigation in SURFEX. The uncertainties issue is also tackled as we discuss about the measurement uncertainties on LAI and LE but also about the remote sensing acquisition uncertainties. The uncertainty related to the model is hard to assess but the local comparison gave us some clue about satisfying performances on the LE simulation outside of irrigation periods. The limitations of the work are also discussed. Especially, we present the limitations introduced by the cloud coverage and the revisit frequency. Finally, the perspectives on the hydrological routing and the introduction of an irrigation process are presented.

**Anonymous Referee #2:** *"English : English must be carefully edited, I provide some examples here page 2 line 4: "vegetation cover present"! "present vegetation cover" page 2 line 5 " the more accurate"! " more accurate" page 2 line 6 "to improving"! "critical to improve" page 3, line 10 "rather than"! "instead of " check in the document the use of "the" page 3, line 9: vegetation type and LAI Page 4, line 4-5: "the ISBA"! "ISBA", Shorter sentences are needed"*

Authors: We have corrected all these examples. We also shortened some sentences in the paper. We particularly paid attention to this in the sections that were edited or added.

**Anonymous Referee #2:** *"the title is too long, some suggestions: use Earth observation instead of remote sensing products use cropland instead of cultivated area use high spatial and temporal resolution"*

Authors:  The title has been modified as suggested.

**Anonymous Referee #2:** *"acronyms must be defined"*

Authors: We have defined all the acronyms that were not already defined.

Again, we greatly appreciate your constructive comments that helped us to improve the manuscript. We hope that we answered to all your concerns in this revision

[revised manuscript text omitted]

---

## Author Response (AR2)

**Referee #1: "In the revised version, the authors addressed and answered appropriately to the comments of my previous review."**

Authors: Thank you again for your comments. They allowed us to clarify the paper.

**Referee #1: "I have still few comments: 1) Fig. 1 is not necessary. Just cite references."**

Authors: As both referees suggested it, this figure has been removed. This is also the case of the figure 2.

**Referee #1: "2) The SURFEX-ISBA model is not described clearly. For instance: a. rows 12-13, why a "single surface temperature is used to compute the different energy fluxes"? the ISBA model should give the possibility to distinguish between bare soil and vegetation types."**

Authors: Thank you for this comment.  As written p4-l33, each vegetation type is simulated separately. A single energy budget is solved on each vegetation type for the whole surface including the soil and vegetation layers. The precision has been added to the sentence:

*A unique energy budget is simulated for each vegetation type on the vegetation-soil layers composite by a single source scheme.*

**Referee #1: "b. The meteorological forcing is drawn from the SAFRAN. Are you not using the meteorological data collected on the field by the eddy covariance towers for driving the model?"**

Authors: Indeed, we could have used the in situ meteorological data as forcing but we choose to use SAFRAN for this study since we aimed at evaluating the added-value of remote sensing data in the perspective of a distributed simulation. For this kind of simulation SAFRAN is the best available forcing (in France). Note that we had previously assessed the SAFRAN data using the meteorological station measurements but did not show the results here since both forcing were very similar in this region. In addition, some meteorological measurements are used as ancillary variables to post-process the eddy covariance fluxes, hence by using SAFRAN we make sure that forcing and validation data are fully independent.

**Referee #1: "3) Need details on the method for land cover maps estimate. For instance, what is the "supervised classification algorithm"?"**

Authors: Some precisions have been added, as well as two references explaining in details the method of classification.

*Then, a supervised classification algorithm based on Iterative Conditional Method (ICM) was applied to determine the vegetation type of each plot (Ducrot et al., 1998, Masse et al., 2011). This algorithm uses a learning sample composed of selected plots where the vegetation type is known. These plots are extracted from the "Politique Agricole Commune" database, made of farmers' land use official declarations. The algorithm then uses the annual Normalized Difference Vegetation Index (NDVI) profiles of these plots to separate all the pixels into 34 classes with similar NDVI profiles.*

**Referee #1: "4) I already did wrote this in my previous review (the answer of the authors don't convince me at all). Why are you not showing the LAI and ET time series for the 2007, 2008, 2009 and 2010 years, and only for 2006? The period 2006-2010 is characterized by interannual variability, such as it is typical in Mediterranean regions. Hence, I would like to see the model performance for all the five years, and see if the model works in the same way for all the years. It ca "**

Authors: The figures for all the years have been added (Fig. 5 and 6 in the new manuscript). However, this analysis made us realize that there is an issue with the evapotranspiration measurements in 2008 and 2010 at the Auradé site (ET is unrealistic in comparison with the other years). This problem is being investigated by the team in charge of the eddy covariance measurements here at CESBIO. They suspect a drift in the air humidity probe. We greatly thank you for this comment which made us realize this problem. The incriminated data and associated scores in table 2 have thus been removed.

The uncertainties on the eddy-covariance measurements are also unavailable yet for years other than 2006 on both sites. For this reason, we would like to keep the graph for the year 2006 as an example to discuss the results and use the graph for 2006-2010 to support the conclusions.

As you can see on the figure 6 of the new manuscript, the delay of the evapotranspiration peak, pointed out while focusing on 2006, is also visible on the year 2008 and 2010 on the Lamasquère site and on the year 2007 on the Auradé site. For these four cases, the crop type over the measurement site was a summer crop. On the Lamasquère site, for 2006 and 2010, the evapotranspiration amplitude is greater on the measurements, reflecting the lack of irrigation in the model over the maize crop. This is not the case for the sunflower crop of 2007 on the Auradé site because there is no irrigation. In 2008 at Lamasquère, the precipitations in spring and at the beginning of the summer were substantial. Hence, the farmer did not irrigate as much as in 2006 and 2010. As a result, the difference of amplitude was less marked. However, the effect of irrigation remains visible at the end of the summer. This analysis shows that the conclusion drawn for the year 2006 remains applicable over the other years. This analysis has been added in the new version of the manuscript.

Page 9:

*Analyzing the results over the entire period (2006-2010) for both sites (Fig. 5 and 6) confirms the behaviors seen on 2006. Indeed, the effect remains small on winter crops (Auradé: 2006 and 2009, Lamasquère: 2007 and 2009) but the delay in the evapotranspiration peak is clearly visible on summer crops (Auradé: 2007, Lamasquère: 2006, 2008 and 2010). The underestimation of the simulations is also visible for the Lamasquère site on 2008 and 2010 but not as marked in 2008. This point will also be discussed in the discussion part (Sect. 5).*

Pages 11-12:

> *The same conclusion applies for 2010 for the Lamasquère site (Fig. 6). Concerning 2008, on Lamasquère too, the precipitations have been sufficient to limit the irrigation. Hence the amplitude of the measured evapotranspiration is lower. Thus the lack of irrigation in the model does not lead to a big difference with the measurements on the maximal amplitude of the evapotranspition, in contrast with 2006 and 2010. Outside of the irrigation period, as well as on rain fed plots, the simulated evapotranspiration when the model is forced by the Formosat-2 appears close to the measured evapotranspiration (Fig. 4 & 6).*

**Referee #2:** *"Review of "Effects of multi-temporal high resolution remote sensing products on simulated hydrometeorological variables in a cultivated area." The authors have made substantial improvements of the paper (content and structure). I provide below remaining issues that need to be addressed prior to final acceptance and publication."*

Authors: Thank you for your comments which are helping us a lot to improve the paper.

**Referee #2:** "*• English and paper structure: - the paper still conveys some redundant sentences, carefully editing the paper is needed prior to publication."*

Authors: We have carefully reviewed the manuscript to identify redundancies. The following sentences were deleted.

Page 9:

> *Local comparison results suggest that the ECOCLIMAP LAI forcing does not allow for a correct representation of the evapotranspiration flux dynamics of summer crops over the entire study area. Indeed it does not capture their phenology with sufficient precision, especially in regard to the temporal extent of the cycle.*

Page 10:

> *Indeed, the peak of the LAI derived from Formosat-2 on a winter wheat field (Fig. 5a) is clearly lower than the measured one but it does not induce a large difference in the evapotranspiration peak (Fig. 6a).*

Page 11:

> *Indeed, for the summer crops, the main differences between the two experiments in terms of LAI are observed during the growth and/or the senescence periods. This is less marked for the winter crops. Indeed, the temporal extent of their phenologic cycle is only slightly modified. These results are confirmed by the spatial analysis (Sect. 4.2).*

**Referee #2:** "*- Avoid the use of "we" in Conclusion, use impersonal statements"*

Authors: The conclusion was modified to avoid the use of "we" or "our".

**Referee #2: "• *Discussion: - I suggest to add subsections to clarify the reading of the discussion*"**

Authors: As suggested, the discussion part has been structured in the following subsections:

5.1 Uncertainties on remote sensing data
5.2 Impact of remote sensing data on simulated evapotranspiration
5.3 Limitations and perspectives

**Referee #2: "- there are still redundancies: e.g. see lines 15 to 20,"**

Authors: As mentioned above, this redundancy has been removed.

> *Indeed, the peak of the LAI derived from Formosat-2 on a winter wheat field (Fig. 5a) is clearly lower than the measured one but it does not induce a large difference in the evapotranspiration peak (Fig. 6a).*

**Referee #2: "- some sentences are not essential: line 22, "the model cannot take it into account""**

Authors: This sentence has been deleted.

**Referee #2: "- conversely to what the authors have announced in response to my comment, there no obvious discussion on the impact of uncertainties in eddy-covariance measurements. This needs to be addressed to strengthen the paper"**

Authors: Following your comment, we have added a part in the discussion section dedicated to the analysis of the eddy-covariance measurements uncertainties:

> *The uncertainties on eddy-covariance measurements (gray area on fig. 4) are calculated from the frequency response correction uncertainty, the Webb-correction (turbulent environment) uncertainty, the calibration correction uncertainty and the random uncertainty, following Kroon et al. (2010). Total uncertainty is proportional to the flux itself and therefore uncertainty grows with the evapotranspiration flux. However these uncertainties remain very small, representing barely 5% of the flux value during summer. Another approach to evaluate the uncertainty is the verification of the energy budget closure. P. Béziat (2009) evaluated the energy budget closure on both sites for the period 2005-2007. His conclusion is that the uncertainty related is very acceptable, the energy budget being closed at around 85-90%. The best results are obtained during the crop cycle. The difference between the evapotranspiration simulated in the ECOCLIMAP experiment and the measurements shows relative error exceeding 100% in spring on summer crops (Fig. 4 & 6). Thus, it exceeds by far the relative uncertainty expected on the eddy-covariance measurements even while taking the energy budget closure uncertainty into account.*

As it was not clear enough in the previous version of the manuscript, the link between the measurements uncertainties and the conclusion on the model bias is showed:

> *The discrepancy between the simulations and the measurments is larger than the observation uncertainties as computed with the method of Kroon et al. (2010, Fig. 4). However, by considering the energy balance closure approach to estimate the observation errors (Beziat, 2009), the difference between the model and the observations is significant only during irrigation periods (Fig. 6).*

**Referee #2: "• Figures: - too many figures: I do not think that Fig1 and 2 are necessary, the authors can refer to the SURFEX references/website for this."**

Authors: As suggested by both referees, the two figures have been removed. The remaining figures have been renamed consequently. The references to figure 1 and 2 have been replaced by ad hoc publications.

**Referee #2: "- The presentation of the Figures need to be revised: - font are too small,"**

Authors: The font size has been increased.

**Referee #2: "- on Fig 8, some acronyms used in the title are not defined (e.g. 'diff') - legend is missing on Fig 8.a"**

Authors: Thank you. The acronym has been defined in the figure caption. The legend was not really missing as it was included in the axis legend. To simplify the understanding of the figure, the legend has been separated from the axis units.

**Referee #2: "• References: - missing background references on the impact of vegetation variables on the simulation of ET: the authors should acknowledge previous published work (recent papers)."**

Authors: Added to the references already present on the influence of LAI on simulated evapotranspiration, we have cited Garrigues et al. (2015). This paper shows the influence of several parameters on the evapotranspiration simulation with SURFEX. Among them, the LAI has a significant impact but the pedotransfer functions seem to have the greatest impact, followed by the presence of irrigation. The pedotransfer functions allow determining the maximum available water content which also depends on the root depth, although the particular impact of the root depth is not quantified in the paper.

This point is discussed in the new manuscript:

*Including irrigation will improve the simulations (Garrigues et al., 2015). An automated irrigation module might therefore be a significant improvement even if it often relies on poorly known soil parameters, like the available water content for evapotranspiration depending on the field capacity, the wilting point and the root depth.*

We remain open to further modify the manuscript if the reviewers wish to suggest additional references.

[revised manuscript text omitted]

L̶o̶c̶a̶l̶ ̶c̶o̶m̶p̶a̶r̶i̶s̶o̶n̶ ̶r̶e̶s̶u̶l̶t̶s̶ ̶s̶u̶g̶g̶e̶s̶t̶ ̶t̶h̶a̶t̶ ̶t̶h̶e̶ ̶E̶C̶O̶C̶L̶I̶M̶A̶P̶ ̶L̶A̶I̶ ̶f̶o̶r̶c̶i̶n̶g̶ ̶d̶o̶e̶s̶ ̶n̶o̶t̶ ̶a̶l̶l̶o̶w̶ ̶f̶o̶r̶ ̶a̶ ̶c̶o̶r̶r̶e̶c̶t̶ ̶r̶e̶p̶r̶e̶s̶e̶n̶t̶a̶t̶i̶o̶n̶ ̶o̶f̶ ̶t̶h̶e̶ e̶v̶a̶p̶o̶t̶r̶a̶n̶s̶p̶i̶r̶a̶t̶i̶o̶n̶ ̶f̶l̶u̶x̶ ̶d̶y̶n̶a̶m̶i̶c̶s̶ ̶o̶f̶ ̶s̶u̶m̶m̶e̶r̶ ̶c̶r̶o̶p̶s̶ ̶o̶v̶e̶r̶ ̶t̶h̶e̶ ̶e̶n̶t̶i̶r̶e̶ ̶s̶t̶u̶d̶y̶ ̶a̶r̶e̶a̶.̶ ̶I̶n̶d̶e̶e̶d̶ ̶i̶t̶ ̶d̶o̶e̶s̶ ̶n̶o̶t̶ ̶c̶a̶p̶t̶u̶r̶e̶ ̶t̶h̶e̶i̶r̶ ̶p̶h̶e̶n̶o̶l̶o̶g̶y̶ ̶w̶i̶t̶h̶

25 s̶u̶f̶f̶i̶c̶i̶e̶n̶t̶ ̶p̶r̶e̶c̶i̶s̶i̶o̶n̶,̶ ̶e̶s̶p̶e̶c̶i̶a̶l̶l̶y̶ ̶i̶n̶ ̶r̶e̶g̶a̶r̶d̶ ̶t̶o̶ ̶t̶h̶e̶ ̶t̶e̶m̶p̶o̶r̶a̶l̶ ̶e̶x̶t̶e̶n̶t̶ ̶o̶f̶ ̶t̶h̶e̶ ̶c̶y̶c̶l̶e̶.̶

[revised manuscript text omitted]

**4 tiles : Sea, Lake, Town and Nature**

**12 different vegetation types (patches)**
multiple patches in a same pixel
→ simulation on each patch and then averaged on the pixel

**ISBA pixel**

**Vegetation types**

1 – Bare soil
2 – Rocks
3 – Permanent snow & ice
4 – Deciduous forest
5 – Evergreen leaves forest
6 – Coniferous forest
7 – C3 crops
8 – C4 crops
9 – Irrigated crops
10 – C3 herbaceous plants
11 – C4 herbaceous plants
12 - Wetland

[Figure]

**Legend**

Study area (Formosat-2 images extent)
Main rivers
Garonne river basin
Border

Figure 1: Study area location (red).

[Figure]

**Figure 42: Land cover map for 2006.**

[revised manuscript text omitted]

---

## Author Response (AR3)

**Editor:**

*"Dear Authors,*

*Thank you for submitting a revised version of this manuscript. Both reviewers thought that most of their comments were addressed satisfactorily and that this version of the manuscript is much improved. Both reviewers also recommended publication after minor revision. Your responses to their second round of the reviews are satisfactory so I am happy to recommend publication of this manuscript after some minor changes (mostly English related).*

*My comments are:*

*(1) P2L26: Suggest adding "of" after suppressing.*
*(2) P3L9: Suggest "consists in" to "consists of".*
*(3) P3L22: Please add "of" after account.*
*(4) P3L23: Please change "firstly" to "first".*
*(5) P3L23: Please change "spatialized" to "spatial".*
*(6) P4L14: "The method is detailed by ...". Please specify which method you are referring to.*
*(7) P7L27: Change "comparison to" to "comparison with".*
*(8) Tables 2 and 3: In the correlation values you are using "," please use "." to indicate decimal values.*
*(9) P8L21: Please change "decrease of around 30% of the RMSE" to "decrease of RMSE by around 30%."*
*(10) P11L6: "However these uncertainties remain very small.." This sentence is repeated twice.*
*(11) P12L6: Please change "Lamasquère, the precipitations have been sufficient" to "Lamasquère, the precipitation amount was sufficient.*
*(12) Figure 8: In the plot titles please change "Differences on" to "Differences in"."*

Authors:

Thank you for your comments. We have corrected the manuscript according to your suggestions. Please find below the marked up version of the manuscript. We thank you and the reviewers again for allowing us to improve the manuscript quality.

[revised manuscript text omitted]

121. Wheat
122. Barley
123. Rapeseed
132. Sunflower
134. Soya
135. Hemp
141. Protein plants
142. Spring barley
1321. Late sunflower
1411. Pea | 7. C3 crops |
| 131. Maize
133. Sorghum
1311. Non-irrigated maize
1312. Silage maize | 8. C4 crops |
| - | 9. Irrigated crops |
| 22. & 112. Fallow
111. Meadow
1111. Temporary meadow
1112. Permanent meadow | 10. C3 herbaceous plants |
| - | 11. C4 herbaceous plants |
| 31. River
32. Lake | 12. Wetland |

**Table 1: Aggregation rules of Formosat-2 cover maps by SURFEX vegetation type**

| Year | Crop type | R² ECOCLIMAP | R² FORMOSAT | RMSE ECOCLIMAP | RMSE FORMOSAT |
|------|-----------|--------------|-------------|----------------|---------------|
| 2006 | Wheat | 0.90 | 0.94 = | 8.07 | 6.86 = |
| 2007 | Sunflower | 0.66 | 0.89 +++ | 20.9 | 12.1+++ |
| 2008 | Wheat | - | - | - | - |
| 2009 | Rapeseed | 0.97 | 0.96 = | 6.71 | 7.66 = |
| 2010 | Wheat | - | - | - | - |

**Table 2: Correlation coefficient and Root-Mean Square Error of evapotranspiration for the Auradé site**

| Year | Crop type | R² ECOCLIMAP | R² FORMOSAT | RMSE ECOCLIMAP | RMSE FORMOSAT |
|------|-----------|--------------|-------------|----------------|---------------|
| 2006 | Maize | 0.15 | 0.65 +++ | 34.4 | 23.5 +++ |
| 2007 | Wheat | 0.76 | 0.95 ++ | 21.9 | 14.6 ++ |
| 2008 | Maize | 0.0.58 | 0.82 +++ | 21.5 | 12.4+++ |
| 2009 | Wheat | 0.94 | 0.95 = | 18.7 | 12.6 ++ |
| 2010 | Maize | 0.46 | 0.62 ++ | 27.2 | 22.8 ++ |

5 **Table 3: Correlation coefficient and Root-Mean Square Error of evapotranspiration for the Lamasquère site**

| Vegetation type | 2006 | 2007 | 2008 | 2009 | 2010 | Interannual mean |
|-----------------|------|------|------|------|------|------------------|
| Outside the crops | +15 (+2.9%) | +16 (+2.3%) | +41 (+5.3%) | +20 (+3.1%) | +30 (+4.7%) | +24 (+3.7%) |
| Wheat | -3 (-0.6%) | -1 (-0.2%) | +61 (+8.0%) | +15 (+2.3%) | +20 (+3.1%) | +18 (+2.8%) |
| Sunflower/soya | +5 (+0.9%) | +54 (+7.9%) | +47 (+6.1%) | +30 (+4.7%) | +48 (+7.5%) | +37 (+5.7%) |
| Maize/sorghum | +4 (+0.7%) | +35 (+5.2%) | +35 (+4.6%) | +18 (+2.8%) | +32 (+5.0%) | +25 (+3.8%) |

**Table 4: Differences between FORMOSAT and ECOCLIMAP experiments on the annual drainage level in mm.yr[-1] and the corresponding fraction of annual precipitations in % (FORMOSAT-ECOCLIMAP).**

| Vegetation type | 2006 | 2007 | 2008 | 2009 | 2010 | Interannual mean |
|---|---|---|---|---|---|---|
| **Outside the crops** | +3 (+0.6%) | +4 (+0.6%) | +7 (+0.9%) | +4 (+0.6%) | +4 (+0.7%) | **+4 (+0.6%)** |
| **Wheat** | +1 (+0.2%) | +5 (+0.7%) | +17 (+2.3%) | +7 (+1.1%) | +4 (+0.6%) | **+7 (+1%)** |
| **Sunflower/soya** | +7 (+1.4%) | +13 (+1.9%) | +10 (+1.4%) | +11 (+1.7%) | +13 (+2.0%) | **+11 (+1.7%)** |
| **Maize/sorghum** | +6 (+1.1%) | +9 (+1.3%) | +10 (+1.3%) | +9 (+1.4%) | +9 (+1.4%) | **+9 (+1.3%)** |

**Table 5: Differences between FORMOSAT and ECOCLIMAP experiments on annual runoff in mm.yr$^{-1}$ and the corresponding fraction of annual precipitations in % (FORMOSAT-ECOCLIMAP).**